# TROPOMI Aerosol Products: Evaluation and Observations of Synoptic Scale Carbonaceous Aerosol Plumes during 2018-2020

Omar Torres[1], Hiren Jethva[2], Changwoo Ahn[3], Glen Jaross[1], and Diego G. Loyola[4]

[1] NASA Goddard Space Flight Center, Greenbelt, MD, 20771, USA

[2] Universities Space Research Association USRA/GESTAR, Columbia, MD, USA

[3] Science Systems and Applications Inc., Lanham, MD USA

[4] German Aerospace Center (DLR), Remote Sensing Technology Institute, Oberpfaffenhofen, 82234 Weßling, Germany

*Correspondence to* Omar Torres (omar.o.torres@nasa.gov)

**Abstract.** TROPOMI near-UV radiances are used as input to an inversion algorithm that simultaneously retrieves aerosol optical depth (AOD) and single scattering albedo (SSA) as well as the qualitative UV Aerosol Index (UVAI). We first present the TROPOMI aerosol algorithm (TropOMAER), an adaptation of the currently operational OMI near-UV (OMAERUV & OMACA) inversion schemes, that takes advantage of TROPOMI's unprecedented fine spatial resolution at UV wavelengths, and the availability of ancillary aerosol-related information to derive aerosol loading in cloud-free and above-cloud aerosols scenes. TROPOMI-retrieved AOD and SSA products are evaluated by direct comparison to sun-photometer measurements. A parallel evaluation analysis of OMAERUV and TropOMAER aerosol products is carried out to separately identify the effect of improved instrument capabilities and algorithm upgrades. Results show TropOMAER improved levels of agreement with respect to those obtained with the heritage coarser-resolution sensor. OMI and TROPOMI aerosol products are also inter-compared at regional daily and monthly temporal scales, as well as globally at monthly and seasonal scales. We then use TropOMAER aerosol retrieval results to discuss the US Northwest and British Columbia 2018 wildfire season, the 2019 biomass burning season in the Amazon Basin, and the unprecedented January 2020 fire season in Australia that injected huge amounts of carbonaceous aerosols in the stratosphere.

## 1 Introduction

The TROPOspheric Monitoring Instrument (TROPOMI) on the Sentinel-5 Precursor (S5P) satellite launched on October 13, 2017 is the first atmospheric monitoring mission within the European Union Copernicus program. The objective of the mission is the operational monitoring of trace gas concentrations for atmospheric chemistry and climate applications. TROPOMI is the follow-on mission to the successful Aura Ozone Monitoring Instrument (OMI, Levelt et al., 2006) that has been operating since October 2004, the Global Ozone Monitoring Experiment-2 (GOME-2, Munro et al., 2016) sensors on the Metop (Meteorological Operational Satellite Program of Europe) satellites operating since 2006, and previous missions such as SCanning Imaging Absorption SpectroMeter for

Atmospheric CHartographY (SCIAMACHY, Bovensmann et al., 1999). The S5P mission precedes the upcoming Sentinel-5 (S5), a TROPOMI-like sensor, and the geostationary Sentinel-4 (S4) missions (Ingmann et al., 2012).

TROPOMI is a high spectral resolution spectrometer covering eight spectral windows from the ultraviolet (UV) to the shortwave infrared (SWIR) regions of the electromagnetic spectrum. The instrument operates in a push-broom configuration, with a swath width of about 2600 km on the Earth's surface. The typical pixel size (near nadir) is 5.5x3.5 $km^2$ for all spectral bands, with the exception of the UV1 (5.5x28 $km^2$) and SWIR (5.5x7 $km^2$) bands. On behalf of the European Space Agency (ESA), the German Aerospace Center (DLR, Deutsches Zentrum für Luft- und Raumfahrt) generates Level 1b calibrated radiance data and level 2 derived products including trace gas ($O_3$, $NO_2$, $SO_2$, CO, $CH_4$, and $CH_2O$), aerosols (UV aerosol index, UVAI), $O_2$-A band aerosol layer height (ALH)) and cloud properties. Currently, no ESA-produced standard quantitative aerosol products are available from TROPOMI. Per an established NASA (National Aeronautics and Space Administration)-ESA interagency collaboration agreement, TROPOMI level 1b calibrated radiance data and level-2 retrieved products, are available at the Goddard Earth Sciences Data and Information Services Center (GES DISC, https://disc.gsfc.nasa.gov/datasets/).

In this paper, we report the first results of a NASA research aerosol algorithm using TROPOMI observations at near-UV wavelengths. TROPOMI aerosol observations will further extend the multi-decadal long near UV aerosol record started with the Total Ozone Mapping Spectrometer (TOMS) series of sensors (1978-1992; 1996-2001, Torres et al., 1998) and continued into the new millennium by the currently operational OMI instrument (Torres et al., 2007). A similar multi-year AOD/SSA record is also available from EPIC (Earth Panchromatic Imaging Camera) on the DSCOVR (Deep Space Climate Observatory) parked at Lagrange point 1 (Marshak et al., 2018; Ahn et al., 2020).

A description of the algorithm is presented in section 2, followed by a detailed evaluation of retrieval results in section 3. In section 4, we use TROPOMI derived information to discuss synoptic-scale aerosol events in different regions of the world since the launch of TROPOMI in 2017.

**2 NASA TROPOMI Aerosol Products**

**2.1 Heritage Algorithm**

The NASA OMI aerosol retrieval algorithms for cloud-free conditions (OMAERUV, Torres et al., 2007; 2013; 2018), and for above-cloud aerosols (OMACA, Torres et al., 2012; Jethva et al., 2018) have been combined into a single algorithm (TropOMAER) and applied to TROPOMI observations. TropOMAER uses observations at two near-UV wavelengths to calculate the UVAI, and to retrieve total column aerosol optical depth (AOD) and single scattering albedo (SSA). Although detailed documentation of the heritage algorithm is available in the published literature, a brief description is presented here for completeness.

*2.1.1 UV Aerosol Index*

TropOMAER ingests measured TROPOMI radiances at 354 nm and 388 nm to calculate the UVAI, a parameter that allows distinguishing UV absorbing particles (carbonaceous and desert dust aerosols, volcanic ash) from non-absorbing particles (Herman et al., 1997; Torres et al., 1998). It is defined as,

$$UVAI = -100log_{10}[I_{354}^{obs}/I_{354}^{cal}] \quad (1),$$

where $I$ represent the observed and calculated radiances at 354 nm. Measurements at 388 nm are used to obtain a wavelength-independent cloud-fraction that is required for the calculation of the $I_{354}^{cal}$ term (Torres et al., 2018). UVAI yields positive values in the presence of absorbing particles, near-zero for clouds, and small negative values for non-absorbing aerosols.

The magnitude of the aerosol UVAI signal depends mainly on AOD, ALH, and aerosol absorption exponent (AAE). For instance, as shown in Fig. 1, for the OMI carbonaceous aerosol model [Torres et al. 2013], and an AAE of 4.8 (i.e., imaginary component of refractive index at 340 nm about 70% higher than at 388 nm), the UVAI increases rapidly with AOD and ALH up to AOD of about 4, at which point the sensitivity to AOD goes down rapidly. For AOD's larger than 6, the UVAI saturates as aerosol absorption of Rayleigh scattered photons peaks, and further UVAI enhancements are only possible for increased values of ALH and/or enhanced aerosol absorption exponent (AAE). Thus, for AOD values larger than about 6, the UVAI effectively becomes a measure of ALH. Although most tropospheric aerosol events fall on the lower left section of Fig. 1 (AOD as large as 4.0 and UVAI as large as 8), observed cases of extraordinarily large UVAI values are generally associated with the injection of huge amounts of UV-absorbing aerosol particles in the upper-troposphere-lower-stratosphere (UTLS) such as ash layers in the aftermath of volcanic eruptions (Krotkov et al., 1999), or wildfire-triggered pyro-cumulonimbus (pyroCb's) episodes (Torres et al., 2020).

The UVAI also contains non-aerosol related information such as ocean color and wavelength-dependent land surface reflectance. It is calculated over the oceans and the continents for all cloud conditions and over ice/snow covered surfaces. TropOMAER UVAI explicitly accounts for the angular scattering effects of water clouds. By doing so the UVAI across-track angular dependence is reduced and spurious non-zero values, produced by the previously used representation of clouds as opaque Lambert Equivalent Reflectors (LER, Torres et al., 2018), are largely eliminated.

*2.1.2 Aerosol Algorithm for cloud-free conditions*

TROPOMI-measured radiances at 354 nm and 388 nm are input into a two-channel inversion algorithm that simultaneously retrieves AOD and SSA for cloud-free conditions (Torres et al., 2007; 2013). Pre-calculated look-up tables (LUTs) of top-of-atmosphere reflectances for pre-defined aerosol types, with nodal points on AOD, SSA and ALH, surface reflectance, and viewing geometry, are used in the inversion process. Ancillary information on surface albedo ALH, and surface type (Torres et al., 2013) is required.

In the inversion algorithm, it is assumed that for each pixel, the aerosol load can be uniquely represented by one of three types: carbonaceous, desert dust or sulfate particles. Each aerosol type is associated with assumed bi-modal

particle size distributions and real component of refractive index (Torres et al., 2007; Jethva and Torres, 2011). Carbonaceous and sulfate particles are assumed to be spherical whereas desert dust aerosols are modelled as non-spherical particles (Torres et al., 2018). UV-absorbing aerosol types are easily differentiated from the non-absorbing kind based on UVAI interpretation. As in the heritage algorithm, observations of carbon monoxide (CO) by AIRS (Atmospheric Infrared Sounder) on the Aqua satellite, are used as a tracer of carbonaceous aerosols to separate them from desert dust particles (Torres et al., 2013).

Because of the known sensitivity of satellite measured UV radiances emanating from UV-absorbing aerosols to ALH (Torres et al., 1998), aerosol layer altitude is prescribed using a combination of a CALIOP (Cloud-Aerosol Lidar with Orthogonal Polarization)-based monthly ALH climatology and transport model calculations (Torres et al, 2013).

For each cloud-free, fully characterized pixel in terms of satellite viewing geometry, surface albedo and type, ALH, and aerosol type, a set of AOD and SSA (388 nm) values is extracted from the LUTs by direct matching to the measured radiances. The aerosol absorption optical depth (AAOD), given by the product of AOD and the single scattering co-albedo (1-SSA), is also reported. In addition to the nominal 388 nm wavelength, parameters are also reported at 354 and 500 nm using the assumed extinction and absorption spectral dependence of the pre-defined aerosol models.

Future algorithm enhancements will explore the utilization of TROPOMI retrieved information on ALH and CO, as well as the additionally available spectral measurements for aerosol typing.

Retrievals are carried out over all ice/snow-free land surface types. Over the oceans, retrievals are made only for pixels characterized by UVAI larger than about 1.0, indicating the clear presence of absorbing aerosols in the atmospheric column. No attempt is made to retrieve properties of weakly absorbing or non-absorbing aerosols over the ocean because of the difficulty in separating the atmospheric aerosol signal from that of ocean color. TropOMAER uses an ESA-produced cloud mask based on sub-kilometer resolution radiance measurements at 1.385 µm by NOAA (National Oceanic and Atmospheric Administration)'s Visible Infrared Imaging Radiometer Suite (VIIRS) on the S-NPP (Suomi-National Polar-orbiting Partnership) platform, re-gridded to the TROPOMI spatial resolution (Siddans, 2016). On March 7, 2020 (TROPOMI orbit 12432), the initial NOAA VIIRS cloud mask used with TROPOMI was replaced with the NOAA Enterprise Cloud Mask (ECM) product. The availability of this product, that facilitates the identification of TROPOMI pixels suitable for aerosol AOD/SSA retrieval, is the only algorithmic improvement of TropOMAER in relation to OMAERUV. The heritage algorithm uses thresholds in measured reflectance, UVAI, and aerosol type [Torres et al., 2013] to identify minimally cloud-contaminated pixels for aerosol retrieval.

*2.1.3 Retrieval of above-cloud aerosol optical depth.*

When absorbing aerosol are present above clouds in overcast conditions, TROPOMI observations at 354 and 388 nm are used to simultaneously retrieve above cloud aerosol optical depth (ACAOD) of carbonaceous or desert

aerosols, as well as the optical depth of the underlying cloud (COD) corrected for aerosol absorption effects Torres et al., 2014).

The algorithmic approach is similar to that of the cloud-free case, except that the retrieved two parameters are ACAOD and COD. Information on single scattering albedo is currently prescribed using an OMI-based long-term SSA climatology (Jethva et al., 2018). The steps involved in aerosol type selection and ALH determination are the same as in the cloud-free retrieval algorithm. A detailed description of the algorithm physical basis and derived products is given in Torres et al. (2014) and Jethva et al., (2018).

**2.2 Calibration**

In this work, we use the UV-VIS (UV/Visible) band 3 of TROPOMI level 1b product (Kleipool et al., 2018). TROPOMI version 1 reflectances for band 3 are within 5%-10% compared with OMI and OMPS (Rozemeijer and Kleipool, 2019). It is expected that the upcoming version 2 of the TROPOMI level 1b product will solve inconsistencies of the radiometric calibration detected in the UV and UVVIS spectrometers using in-flight measurements and it will include degradation correction for the affected bands (Ludewig et al., 2020).

For this application, we use TROPOMI correction coefficients at 354 and 388 nm derived using an ice reflectance based vicarious approach that has been used to evaluate the calibration of UV-VIS sensors (Jaross and Warner, 2008).

TROPOMI measured reflectances over Antarctica on 28 and 29 November 2017 were compared to radiative transfer model results. We calculate the ratio of each observed across-track ground pixel's reflectance at a specified wavelength to that of the modeled value for the same viewing conditions to obtain an error for that measurement. The model used is exactly the same as was used in the generation of OMI Collection 3 level 1b data (Dobber et al., 2008). The static corrections applied to TROPOMI reflectances elsewhere on the globe were derived by first averaging over all measurement errors at a given across-track position, then further smoothing with a 5-pixel boxcar in the across-track direction. Corrections range from -4% to +2% in the across-track direction for the two wavelengths. We plan to repeat the calibration adjustments and to reprocess when an improved version 2 of the level 1b product is released by ESA.

**3 Evaluation TropOMAER Performance**

Improved performance of the TropOMAER algorithm in relation to the OMI heritage algorithm is expected as a consequence of both instrumental and algorithmic enhancements. TROPOMI 5.5x3.5 km$^2$ spatial resolution represents a factor of 16 improvement in relation to OMI's 13x24 km. In addition to its finer nadir resolution, TROPOMI's extreme off-nadir resolution does not increase as much as OMI's. As discussed in section 2.1, the TROPOMI-dedicated VIIRS cloud mask is the only algorithmic improvement in the current version of TropOMAER.

In this section, we evaluate TropOMAER UVAI product in relation to its OMAERUV predecessor, and also compare it to the operational ESA/KNMI (Koninklijk Nerderlands Meteorogisch Instituut) TROPOMI UVAI product (Stein, 2018). We also evaluate the accuracy of TROPOMI quantitative AOD and SSA aerosol products by

comparison to ground-based independent observations. TROPOMI derived aerosol parameters are also compared to OMI results during the same time and similar regions.

**3.1 UV Aerosol Index Evaluation**

Two consecutive orbit views by OMI and TROPOMI of the smoke plume from the Pacific Northwest fires on August 18, 2018 are shown in Figure 2. OMI's depiction of this event appears in Fig. 2a whereas Fig. 2b illustrates the same aerosol feature as reported by the TropOMAER algorithm. Both products cover a similar range of UVAI values from a slightly negative background to values as high as 10. OMI's coarse spatial resolution, however, is in stark contrast to TROPOMI's fine resolution that allows the mapping of the smoke plume UVAI signal with unprecedented level of detail. Missing data in OMI's depiction in Fig. 2a, is associated with the row anomaly that has reduced the sensor's observing capability by nearly 50% since about 2008 (Torres et al., 2018; Schenkeveld, Jaross at al., 2017). Figure 2c, shows the operational TROPOMI ESA/KNMI UVAI product for the same event. The main difference between the NASA (Fig. 2b) and ESA/KNMI (Fig. 2c) UVAI products is the background values that, while near-zero for the NASA product, reaches values a low as -2 for the KNMI product. The large background difference between the two products is likely the combined effect of calibration uncertainties in the operational ESA/KNMI product, as well as algorithmic differences in the treatment of clouds in the calculated component of the UVAI definition. In the KNMI UVAI calculation, clouds are modelled as opaque reflectors at the ground (Herman et al., 1997), whereas in the NASA UVAI, clouds are explicitly modelled as poly-dispersions of liquid water droplets using Mie Theory (Torres et al., 2018). A comparative analysis of OMAERUV and TropOMAER UVAI is presented in section 3.3.

**3.2 Evaluation of retrieved Aerosol Optical Depth and Single Scattering Albedo**

We evaluate separately the effect of instrumental and algorithmic improvements in TropOMAER retrieval algorithm by direct comparison of the satellite product to ground-based globally distributed (over land) level 2 Version 3 measurements of AOD (Giles et al., 2019) by the Aerosol Robotic Network (AERONET, Holben et al., 1998).

Measurements of AOD at 380 nm are available at most AERONET sites, allowing a direct comparison to OMI and TROPOMI 388 nm retrievals. No attempt was made to account for the small AERONET-TROPOMI wavelength difference. AERONET AOD measurements at the twelve sites listed in Table 1 over a two-year period (May-2018 thru May 2020) were used in the analysis. These locations were chosen based on the availability of 380 nm AOD measurements, and on the representativity of environments where most common aerosol types (carbonaceous, desert dust, and sulfate-based) are observed.

*3.2.1 Impact of TROPOMI's fine resolution on AOD retrieval*

We first analyze the impact of the enhanced spatial resolution by independently comparing OMI retrievals by the OMAERUV algorithm and TropOMAER AOD inversions to AERONET measurements over the selected set of AERONET sites. In this validation exercise, the VIIRS cloud mask is ignored, and the heritage algorithm cloud mask [Torres et al., 2013] is applied to both OMI and TROPOMI observations. Resulting relevant statistics for the

two validations were compared. These stastistics based on an admittedly small sample of observations, are only intended to illustrate the relative improvement in the accuracy of retrieved parameters associated with TROPOMI enhanced instrumental and algorithmic capabilities with respect to OMI. This is by no means an exhaustive validation exercise of the TROPOMI record for which a lot more AERONET observations are needed.

Ground-based AOD values averaged within ±10 min of the satellite overpass, are compared to spatially averaged retrievals by OMAERUV within a 40 km radius, and by TropOMAER within 20 km (because of the smaller pixel size) of the AERONET site. Figure 3 shows scatter plots of the AERONET-satellite comparisons at the combined 12 sites for OMAERUV (Fig. 3a) and TropOMAER (Fig 3b). The dotted envelope lines indicate the calculated expected uncertainty of retrieved AOD (larger of 0.1 or 30%) associated with uncertainties in assumed ALH and cloud contamination (Torres et al., 1998; 2007). The calculated  relevant statistics are listed in columns 2 and 3 of Table 2. The TROPOMI-AERONET comparison yields 741 matchups compared to OMI's 410, representing an 80% increase. The larger number of coincidences is the result of the combined effect of  TROPOMI's finer spatial resolution as well as the OMI's row anomaly (Torres et al., 2018; Schenkeveld, Jaross et al., 2017) affecting OMI since 2007. In spite of a large number of outliers in the lower AOD range (up to about 0.7) coming from a few sites (see section 3.2.2), the TROPOMI-AERONET comparison in Fig. 3b   yields an improved correlation coefficient (0.82) with  respect to the one (0.60) associated with the OMI observations. The lowest OMAERUV reported correlation coefficients are associated with outlying large AOD estimates resulting from mixtures of UV-absorbing aerosols and clouds, which are difficult to identify at OMAERUV's coarse spatial resolution. Resulting root mean square errors (rmse) values are 0.31 and 0.19 for OMI and TROPOMI, respectively. The reported statistics suggest a clear performance improvement of the TROPOMI algorithm directly linked to the sensor's smaller pixel size.

*3.2.2 Effect of VIIRS cloud masking on AOD retrieval*

The effect of using the VIIRS cloud mask re-gridded to the S5P resolution (Siddans et al., 2016) to identify cloud-free pixels was evaluated by means of a third validation exercise. This time, the TROPOMI-AERONET comparison was carried out for an enhanced TropOMAER algorithm that makes use of the VIIRS dedicated cloud mask.  The scatter plot illustrating the outcome of the later comparison is shown in Figure 3c.  The corresponding correlation coefficient and root mean square errors  are listed in column 4 of Table 2. An inspection of columns 3 and 4, shows that using the VIIRS cloud mask translates into an increase in the number of matchups of over 100 (to 845) as well as higher correlation coefficient (0.89)   and slightly improved rmse (0.16) value than that reported for the TropOMAER algorithm with heritage cloud mask. A slightly reduced number of  TROPOMI AOD outliers in the 0 to 0.5 range are still observed in Fig. 3c. A close examination of the source of those points indicate that most of them come from likely cloud contaminated observations at the Banizoumbou, Beijing and Mongu sites (shown in the scattered plots for each of the 12 sites in the analysis shown in Appendix A) where carbonaceous aerosols and sub-pixel size clouds co-exist, making cloud screening a particularly difficult task.

*3.2.3 SSA Evaluation*

An analysis similar to that carried out for AOD evaluation is performed for SSA using AERONET Version 3, level

2 inversion product (Sinyuk et al., 2020). The AERONET inversion algorithm that infers aerosol particle size

distribution and complex refractive index (from which SSA is calculated) does not include measured sky radiances

nor retrieved AOD at wavelengths shorter than 440 nm. Therefore, the evaluation of OMI and TROPOMI retrieved

388 nm SSA requires a wavelength transformation of the satellite products to 440 nm based on the assumed spectral

dependence of absorption for each aerosol type in the algorithm (Jethva et al., 2014). Unlike in the AOD validation,

in which the AERONET observation is considered a ground-truth measurement, the AERONET SSA product is the

result of a remote sensing inversion and, just like the satellite retrievals, subject to non-unique solutions. Thus, the

AERONET-satellite SSA analyses discussed here cannot be regarded as a validation of the satellite product,  but

merely a comparison of the outcome of two independent inversion methods.

Since AERONET's retrieved SSA is accurate within 0.03 for 440 nm AOD ≥ 0.4 (Dubovik et al., 2002, Sinyuk et

al., 2020), observations at many sites are required to get meaningful statistics. Thus, OMI and TROPOMI SSA

retrievals were averaged in a grid box of size 0.5 deg. x 0.5 deg. centered at the AERONET station at 164 sites.

Because AERONET SSA derived from almucantar scans is considered unreliable at near noon (Dubovik et al.,

2002) when satellite overpass occurs, the AERONET Level-2 SSA data were averaged within a ±3 hour window

from the TROPOMI overpass time under the implicit (and admittedly untested) assumption that SSA does not vary

significantly throughout the day. The chosen six-hour temporal window allows early morning and late afternoon

inversions that are expected to have better accuracy due to larger solar zenith angle and longer atmospheric path

length. Although the Version 3 AERONET product has recently introduced hybrid scans aimed at sampling larger

air masses covering a wider range scattering angles during the middle of the day, only a fraction of currently

deployed sensors are capable of such measurements (Sinyuk et al., 2020).

Similarly to the previously described AOD validation exercise, satellite-AERONET SSA comparisons were made

by independently applying the heritage cloud screening to OMAERUV retrievals and, both heritage and VIIRS-

based cloud masking approaches, to TropOMAER. Figure 4 displays the results of the comparison for different

aerosol types. The AERONET-OMI analysis is shown in Fig. 4a, and the result of the AERONET-TROPOMI

comparison using heritage cloud screening is displayed in Fig. 4b, whereas the outcome when using the VIIRS cloud

mask in the TROPOMI inversion appears in Fig. 4c. A numerical summary of the results is presented in Table 3. In

a similar fashion as observed in the AOD retrieval evaluation,  the number of coincidences increases from 303 for

OMI to 323 for TROPOMI with heritage cloud screening, and to 415 for the TROPOMI/VIIRS cloud mask

combination. The reported root-mean-square-difference (rmsd) between the two measurements varies little between

the three comparisons. The percent number of retrievals within the stated uncertainty levels is marginally better for

OMI than TROPOMI with heritage cloud screening, and significantly better for OMI than TROPOMI with VIIRS

cloud mask. A visual inspection of Fig. 4 shows that the satellite retrieved SSA for dust is overestimated for

AERONET SSA values lower than about 0.9 in the three comparisons. The observed apparent overestimation of the

satellite SSA values for desert dust aerosols (blue symbols) in the OMI comparisons (Figure 4a) has been previously

observed and discussed in the literature (Jethva et al., 2014). The apparent overestimation shown in the TROPOMI

results (Figs, 4b and 4c) are discernibly larger than seen in the OMI data (Fig 4a). Figs. 4b and 4c also show a clear

overestimate in the retrieved SSA of smoke aerosols (red symbols) not seen in the OMI retrievals in Fig. 4a. In

general, for all three aerosol types, TROPOMI SSA retrievals are seemingly biased high by 0.01-0.02 compared to

those from OMI, suggesting a possible connection with remaining TROPOMI L1 calibration issues.

**3.3 OMI-TROPOMI long term continuity**

The continuity of the OMI and TROPOMI records of aerosol properties is analyzed in this section. Monthly average

values of AOD and AAOD for May 2018 to May 2020 two-year period, calculated for three regions: Eastern United

States (EUS) between 25–45◦N and 60– 90◦W; southern Africa (SAF), bounded by 5–25◦S and 15– 35◦E and the

Sahara Desert (SAH) zone between 15–30◦N and 30◦E–10◦W. The EUS region is representative of areas

predominantly associated with non-absorbing aerosols and clouds. The SAF region is known as an important source

area of carbonaceous aerosol-cloud mixtures, whereas the SAH region is the source area of the desert dust part, the

most abundant aerosol type.

Figure 5 shows the two-year AOD record produced by the OMAERUV (blue) and TropOMAER (red) algorithms

for the three regions. TropOMAER-generated AOD values are consistently higher by about 0.2 than the

OMAERUV record for the SAF and SAH regions where the absorbing aerosol load is typically large most of the

16 year. The EUS region shows significantly smaller OMI-TROPOMI differences in monthly mean values. The

17 comparison was also done using a TropOMAER version of the algorithm that uses the heritage cloud screening

approach, yielding similar results.

Figure 6 depicts the two-year record in terms of AAOD. Differences as large as 0.03 in the SAH region during the

2018 Spring-Summer months are significantly lower in the 2019 record. Overall, the AAOD time series over the

three regions show closer agreement between the two sensors, suggesting a partial cancellation of retrieval errors in

SSA and AOD when combined in the AAOD parameter.

Figure 7 shows global three-month (June, July, August 2018) average maps of AAOD from TROPOMI (top) and

OMI (bottom) observations. Seasonally occurring features such as the Saharan desert dust signal over Northern

Africa and the smoke plumes associated with biomass burning over Namibia, Angola, and Congo are clearly picked

by both sensors with comparable AAOD values. Other continental aerosol features such as dust and smoke signal

over the western US, and smoke plumes from wildfires in the Norwest Pacific and moving eastward across Canada

are detected at similar AAOD values by the two sensors, albeit with a higher level of detail in the TROPOMI

product. Similar aerosol signals are also picked up by the two sensors over Saudi Arabia, norwest India, Pakistan,

and Western China. Perhaps, the most striking continental difference in the seasonal map in Fig. 7 is the much larger

OMI background AAOD in South America, possibly linked to the difficulty of removing sub-pixel cloud effects at

OMI's resolution.

Surprisingly, OMI only shows a very scattered signal of the North Atlantic Saharan dust plume between Northern

Africa and the plume's leading edge north of Venezuela over the Caribbean, whereas the TROPOMI product shows

an almost continuous North Atlantic plume. In spite of the geographically sparse nature of the OMI AAOD data,

there is high consistency in the retrieved values by the two sensors. A similar but less severe difference is also

observed over the South Atlantic, where the OMI retrieved carbonaceous aerosol plume is more disperse than what

is shown in the TROPOMI map. The combined effect of prevailing sub-pixel cloud contamination and OMI's row anomaly explains the spatially scattered OMI retrievals over the oceans.

Clearly, the full TROPOMI coverage at much higher spatial resolution than OMI and the high-resolution VIIRS cloud mask contribute to significantly improve the near UV aerosol product.

The OMI and TROPOMI gridded 2018 monthly data used to produce the seasonal average maps discussed above are also displayed in Figure 8 as density AAOD (left) and UVAI (right) plots. Although small offsets in UVAI (~0.2) and AAOD (~0.02) between the sensors are apparent, a high degree of correlation between the observations by the two instruments is clearly observed.

## 4 TROPOMI  view of Important Aerosol Events

In this section, we briefly discuss three major continental scale aerosol events that took place during the two-year period following the operational implementation of the S5P mission. The discussed cases include the occurrence of wildfire plumes in both hemispheres, while the third one is likely associated with agricultural practices involving biomass burning in the Amazon region.

### 4.1 2018 Fire Season in Northwest USA and Canadian British Columbia

The 2018 fire season in the western USA and Canadian British Columbia territory was one of the most active of the last few years. It is estimated that over 8500 fires were responsible for the burning of over 0.8 million hectares, which is the largest area burned ever recorded according to the California Department of Forestry and Fire Protection (fire.ca.gov) and the National Interagency Fire Center (nfic.gov). From mid-July to August, intense fires in Northern California, including the destructive Carr and Mendocino Complex fires, produced elevated smoke layers that drifted to the east and northeast. In 2018, the British Columbia (BC) province of Canada encountered its worst fire season on record, surpassing the 2017 record, with more than 2000 wildfires and 1.55 million hectares burned accounting for about 60% of the total burned area in Canada in 2018 (https://www2.gov.bc.ca/gov/content/safety/wildfire-status).  Figure 9 shows the spatial extent of the smoke plume generated by wildfires in Canadian B.C. and northwestern USA on August 18, in terms of the 388 nm AOD, and SSA products from both TROPOMI (top) and OMI (bottom) observations (the corresponding UVAI depiction was shown in Fig. 2). Observed gaps in the core of the plume are due to out of bounds retrieval conditions. The carbonaceous aerosol layers produced by the fires spread over a huge area covering large regions of USA's Midwest and Central Canada. The height of the aerosol layer varies between 3 and 5 km according to CALIOP observations (not shown). Although OMI's coarse resolution and row-anomaly related reduced spatial coverage are clearly observable, the retrieved AOD and SSA fields by the two sensors look remarkably similar. TROPOMI and OMI AOD retrievals reach values as high as 5.0 near the sources, generally consistent with AERONET ground-based observations that, on this day, reported AOD values as large as 1.5 (412 nm) at the Lake Erie site (41.8ºN, 83.2ºW) and values in excess of 3.0 at the Toronto station (43.8ºN, 79.5ºW). SSA values in the range 0.85-0.92 are retrieved by both sensors over the extended area. Minimum  OMI retrieved SSA (0.85) in the vicinity of a source area,

however, is lower by about 0.02 than the corresponding TROPOMI measurement, consistent with the relative OMI-TROPOMI SSA differences reported in Fig. 4.

**4.2 Amazon Basin 2019 Fires**

Figure 10 shows the spatial distribution of the September 2019 average TROPOMI UVAI, AOD and AAOD over the region between the Equator and 40ºS and between 35ºW and 85ºW. Monthly average AOD values of around 2.0 prevailed over the source areas. The smoke plumes were mobilized downwind towards southern Brazil reaching highly populated areas, where TROPOMI-measured monthly average AOD in the range 0.9 to 1.0 are reported.

Figure 11 shows the time series of monthly average OMI 388nm AOD over the region for the last 15 years, along with the overlapping TROPOMI AOD observations over the last two years, illustrating the importance of the continuity of the longterm record. Although, as discussed earlier, there are small differences in the time series between the two sensors, these differences are not large enough to question the ability to recognize years with large seasonal events from years with comparably reduced biomass burning activity. Seasonal carbonaceous aerosol concentration over the Amazon Basin associated with intense agriculture-related biomass burning has significantly decreased over the last twelve years since 2008. The OMI record shows a remarkable decrease since 2008 when near record high values were observed (Torres et al., 2010). After consecutive AOD September peaks larger than 2.0, in the three-year 2005-2007 period, the monthly average AOD over the Amazon basin reduced to values about 0.5. An isolated abrupt increase to larger than 2.0 was again observed in 2010. Since then, the September peak AOD value has remained much lower than 1, except for 2017 and 2019, when September average AOD larger than unity was observed. The 2019 peak AOD value (1.25) was also retrieved by TROPOMI observations. Although the overall regional average was slighter larger than in the previous year, it was about a third of the 2010 peak value. As a result of the prevailing regional atmospheric dynamics in 2019, carbonaceous aerosols generated by seasonal biomass burning over regions up north were transported towards the southeast, reaching large urban centers such as Sao Paulo and Curitiba, generating a lot of media attention (Hughes, 2019).

**4.3 Australia 2019-2020 Fires**

The 2019-2020 fire season in Australia resulted in 18.6 million burned hectares, most of them in the New South Wales and Victoria southeastern states (SBS News, 2020). It is estimated tens of people died along with billions of animals that were exterminated, including species that were near extinction before the fire (Readfearn, 2020). The intense fire activity likely triggered a number of pyroCb clouds over a few days between December 30, 2019 and early January 2020, injected large amounts of carbonaceous aerosols into the Southern Hemisphere UTLS (Ohneiser et al., 2020). In this section, we describe TROPOMI observations of these events in terms of UVAI and AOD retrievals. As observed in visible satellite imagery (not shown) most of the UTLS injected carbonaceous aerosol material was initially above clouds. TROPOMI near UV observations were used in conjunction with aerosol layer height from CALIOP observations as input to a modified version of the TROPOMI aerosol algorithm that handles stratospheric aerosol layers (TropOMAER-UTLS). The retrieved SSA over clear scenes was then used as input in the retrieval of AOD over cloudy pixels by the above-cloud-aerosol module described in section 2.1.3.

TROPOMI retrieved AOD was used to produce an estimate of resulting stratospheric aerosol mass (SAM). The SAM calculation procedure involves the separation of the stratospheric AOD component from the total AOD column measurement, and the use of an extinction-to-mass-conversion approximation described in Appendix A. This approach was previously applied to EPIC near UV AOD retrievals to calculate the SAM associated with the 2017 British Columbia pyroCb's events (Torres et al., 2020).

The identification of stratospheric aerosols is carried out establishing a theoretical relationship between AOD and UVAI for a hypothetical aerosol layer at the tropopause for assumed values of ALH and AAE (see discussion in section 2.1). CALIOP provided ALH information, and assumed AAE value of 4.8 similar to that in Torres et al (2020) were used as input to TropOMAER-UTLS. AOD retrievals associated with UVAI values larger than those indicated by the AOD-UVAI relationship at the tropopause height are assumed to correspond to stratospheric aerosols. Figure 12 shows TROPOMI observed UVAI (y-axis) and retrieved AOD (x-axis) for CALIOP-reported ALH on December 31, 2019. Data points in red indicate retrieval lying above the estimated tropopause height (12 km), while the blue points show retrievals at heights below that level. The altitude locations of the retrievals in relation to the tropopause are determined based on unique viewing-geometry-dependent UVAI-AOD relation for each pixel, difficult to visualize on a single plot. Therefore, a quadratic fit (black line) to all data, i.e., above and below the tropopause, was derived to illustrate, for visualization purposes, the separation of tropospheric and stratospheric aerosols.

Unlike during the 2017 British Columbia fire episodes, when a large fraction of the pyroCb generated aerosol plume remained initially in the troposphere and some of it ascended diabatically to the stratosphere over the next few days (Torres et al., 2020), during the Australian 2020 pyro-convective fires most of the produced carbonaceous aerosols appear to have gone directly into the stratosphere. Figure 13 shows TROPOMI retrieved UVAI and AOD fields (total column and stratospheric component) on January 2, 2020. Only small differences in the total column and above-tropopause AOD fields are observed, as most of the aerosol material was directly deposited in the stratosphere.

Stratospheric AOD values were converted to mass estimates using the procedure described in Torres et al. (2020) and also included as Appendix B of this paper. For mass estimation purposes, TropOMAER 388 nm AOD data was gridded to 0.25°x0.25° lat.-lon. resolution. Figure 14 shows calculated daily SAM values (in kilotons) from December 31, 2019 thru January 7, 2020, resulting from aerosols above 12 km, altitude used as a proxy of the tropopause height. Separate aerosol mass retrievals were carried out for cloud-free (blue bars) and cloudy scenes (green bars), with the daily total SAM given as the sum of these two components (orange bars). The observed daily monotonic increase from 119 kt on December 31, 2019 to 380 kt on January 2, 2020 is likely the result of distinct pyroCb events that seemingly injected most of the aerosol mass directly in the stratosphere. Following the January 2 maximum, SAM decreases over the following three days to a minimum of 87 kt on January 5, likely due to the combined effect of dilution processes, that spread the aerosol layer horizontally and thins it out to extremely low AOD values beyond the sensor's sensitivity to the total AOD column measurement, as well as aerosol deposition bringing it down to lower than 12 km and, therefore, no longer included in the SAM calculation.

The sudden increase to 166 kt on January 6 is likely associated with another pyroCb event observed on January 4 that injects an additional 166 kt. Thus, the TROPOMI-based total SAM estimate is the sum of the two peaks on January 2 and January 6 yielding a total of 546 kt, which about twice as much as the 268 kt estimated SAM for the 2017 British Columbia pyroCb [Torres et al., 2020] using the same mass estimation technique. The uncertainty of the estimated SAM is ±40%, which represents the combined effect of uncertainties on assumed AAE (±0.5) in the AOD retrieval, and the uncertainty associated with the assumed aerosol density range of 0.79 and 1.53 g-cm−3 (Reid et al., 2005).

**5 Summary and future work**

The NASA TropOMAER aerosol algorithm applied to TROPOMI observations is an adapted version of the OMAERUV algorithm developed for OMI. Currently, the only algorithm upgrade of TropOMAER is the use of a dedicated VIIRS-based cloud mask. Initial retrieval results for the first two years of operation of the TROPOMI sensor were reported.

Since radiometric calibration uncertainties in the range 5-10%, relative to OMI and S-NPP OMPS measurements, are reportedly present the TROPOMI version 1 level1b UVVIS (UV/Visible) band 3 (Rozemeijer and Kleipool, 2019), we applied vicariously derived correction factors to TROPOMI measured radiances at 354 and 388 nm. The approach, based on measured ice reflectances and radiative transfer calculations, yield corrections in the range from -4% to +2% in the across-track direction for both wavelengths.

The AERONET Version 3, level 2 380 nm AOD data record was used to evaluate the performance of the TropOMAER algorithm. An AERONET AOD data aggregate consisting of two years (May 2018-May 2020) of observations at 12 sites representative of most commonly aerosol types (i.e., carbonaceous, desert dust, and urban-industrial aerosols) was used in the analysis. To separately evaluate the effects of instrumental and algorithmic improvements on retrieved products, we carried out a three-way comparison of satellite retrieved AOD to AERONET observations: 1) OMI retrievals by the OMAERUV algorithm, 2) TropOMAER retrievals using the heritage (OMAERUV) cloud screening method, and 3) TropOMAER retrievals using a VIIRS-based cloud mask were independently compared to AERONET observations. A comparative analysis of evaluations 1 and 2 shows the impact of enhanced instrumental capabilities, whereas the analysis of evaluations 2 and 3 highlights the effect of using the VIIRS cloud mask.

Results from comparisons 1 and 2 indicate that a large increase in the number of matched observations (from 410 to 741) and higher correlation coefficient (from 0.60 to 0.82) are the main benefit of TROPOMI's enhanced resolution. Resulting rmse values are similar for both comparisons. The comparison of evaluations 2 and 3, intended to evaluate benefits associated with the availability VIIRS cloud mask, shows an additional increase in the number of matched pairs (from 741 to 845) and higher correlation coefficient (from 0.82 to 0.89). The multi-site AERONET-TROPOMI analysis shows the presence of over-estimated AOD values in the 0 to 0.5 range. The presence of these outliers is not a common feature at all sites but primarily associated with the presence of carbonaceous aerosols and cloud mixtures that the current cloud masking scheme apparently fails to identify. Future work to improve the

current cloud masking approach is planned. A similar analysis using observations at 164 sites was carried out to evaluate TROPOMI's SSA product yielding the similar main conclusion of increased number or retrieval opportunities for the higher spatial resolution sensor.

The observed improvement associated with TROPOMI's higher spatial resolution and, therefore, increased number of retrieval opportunities compared to OMI, may be over-estimated  in view of the row anomaly affecting the OMI sensor that has reduced by nearly 50% its viewing capability.

The TropOMAER aerosol products were also evaluated by direct comparison to OMI at daily, monthly, and seasonal temporal scales. A comparative analysis OMI and TROPOMI two-year time series of 388 nm AOD monthly values shows that TROPOMI AOD values are higher than OMI by about 0.2. This AOD offset is of about the same magnitude as identified in the validation analysis using AERONET observations.

Although TROPOMI products show improved spatial coverage especially over the oceans where clouds are a significant obstacle at OMI's coarse resolution, the reported comparisons show an overall consistent picture that allows for the long-term continuity of the near-UV aerosol record.

Three continental-scale carbonaceous aerosol events over the last two years captured the attention of climate scientists and news media alike. These events, observed by TROPOMI, were briefly described here in terms of TropOMAER products.

The atmospheric aerosol load generated by the hundreds of fires in the western USA and Southern Canada in the summer of 2018 was measured by both ground-based and spaceborne sensors. The fires-triggered aerosol layers extended over a huge area covering large regions of the USA's Midwest and Central Canada. Except for the difference in spatial resolution, OMI and TROPOMI observations yield a consistent view of this event with  UVAI values as large as 10 produced and retrieved AOD values as high as 5.0, consistent with AERONET ground based observations at several sites.

After eight years of noticeable reduced biomass burning in Southern Brazil during August and September, high levels of carbonaceous aerosols were detected in 2019 by both OMI and TROPOMI. As a result of prevailing regional atmospheric dynamics in 2019, carbonaceous aerosols generated by seasonal biomass burning were transported towards the southeast reaching large urban centers. OMI and TROPOMI reported September 2019 monthly and regional average AOD was slightly larger than in the previous year, and about a third of OMI reported 2010 peak (~2.5) value.

A number of pyroCb's likely triggered by intense bushfires in the New South Wales province of Australia between December 30, 2019 and early January 2020 injected large amounts of carbonaceous aerosols into the Southern Hemisphere UTLS. Very large values of TROPOMI UVAI observations pointed to an elevated aerosol layer, which was confirmed by CALIOP reports of a distinct high-altitude aerosol layer near 12 km, above tropospheric clouds. TROPOMI-retrieved AOD over both cloud-free and cloudy scenes was used to produce an estimate of the injected

aerosol mass above 12 km, yielding a total of 546 kt, which is at least twice as much as the estimated carbonaceous

aerosol mass injected into the stratosphere by the 2017 Canadian fires.

Future TropOMAER algorithm enhancement will explore the utilization of TROPOMI retrieved information on

aerosol layer height (Nanda et al., 2019), CO (Martínez-Alonso et al., 2020), clouds (Loyola et al., 2018), geometry-

dependent effective LER (Loyola et al., 2020), as well as taking advantage of additional available spectral

measurements for aerosol typing. Work is currently underway on the development of a higher spatial resolution

surface albedo data and on the optimization of the instrument characterization.

*Author contributions.* The leading author conceptualized the study and wrote the paper. Co-authors HJ, CA, and DL contributed to the data analysis in the manuscript. Co-author GJ contributed the vicarious instrumental calibration work used in the interpretation of the satellite observations.

*Competing interests.* The authors declare that they have no conflict of interest.

*Acknowledgements.*
Thanks are due to the anonymous reviewers whose constructive feedback led to significant improvement of the article.

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

| Site (country) | Lat., Lon. |
|---|---|
| Hohenpeissenberg (Germany) | 47.8ºN, 11.0ºE |
| GSFC (USA) | 39.0ºN, 76.8ºW |
| Lille (France) | 50.6ºN, 3.1ºE |
| Beijing-CAMS (China) | 39.9ºN, 116.3ºE |
| Thessaloniki (Greece) | 40.6ºN, 23.0ºE |
| Fukuoka (Japan) | 33.5ºN, 130.5ºE |
| Banizoumbou (Niger) | 13.5ºN, 2.7ºE |
| Mongu (Zambia) | 15.3ºS, 23.3ºE |
| Leipzig (Germany) | 51.4ºN, 12.4ºE |
| Lumbini (Nepal) | 27.5ºN, 83.3ºE |
| Yonsei_University (S. Korea) | 37.6ºN, 126.9ºE |
| New Delhi (India) | 28.6ºN, 77.2ºE |

3 **Table 1: AERONET sites used for the AOD validation analysis presented in this study.**

|  | OMAERUV | TropOMAER (Heritage Cloud Mask) | TropOMAER (VIIRS Cloud Mask) |
|---|---|---|---|
| Number of matchups | 410 | 741 | 845 |
| Correlation coefficient | 0.62 | 0.82 | 0.89 |
| Root Mean Square | 0.31 | 0.19 | 0.16 |

1    Table 2. Summary of linear fit results between AERONET measured and satellite retrieved AOD at 12 locations

2    (column 1) by the OMAERUV algorithm (column 2), TropOMAER Heritage algorithm (column 3), and

3    TropOMAER algorithm with VIIRS cloud mask (column 4).

|  | OMAERUV | TropOMAER (Heritage Cloud Mask) | TropOMAER (VIIRS Cloud Mask) |
|---|---|---|---|
| Number of matchups | 303 | 323 | 415 |
| Root Mean Square | 0.046 | 0.040 | 0.044 |
| Percent within 0.03 | 52 | 51 | 48 |
| Percent within 0.05 | 78 | 75 | 70 |

5    **Table 3.** Number of coincidences, root mean square, and percent number of SSA retrievals within 0.03 and 0.05 of

6    AERONET values (column 1)  for OMAERUV (column 2), TropOMAER with heritage cloud mask, and

7    TropOMAER with VIIRS cloud mask (column 3).

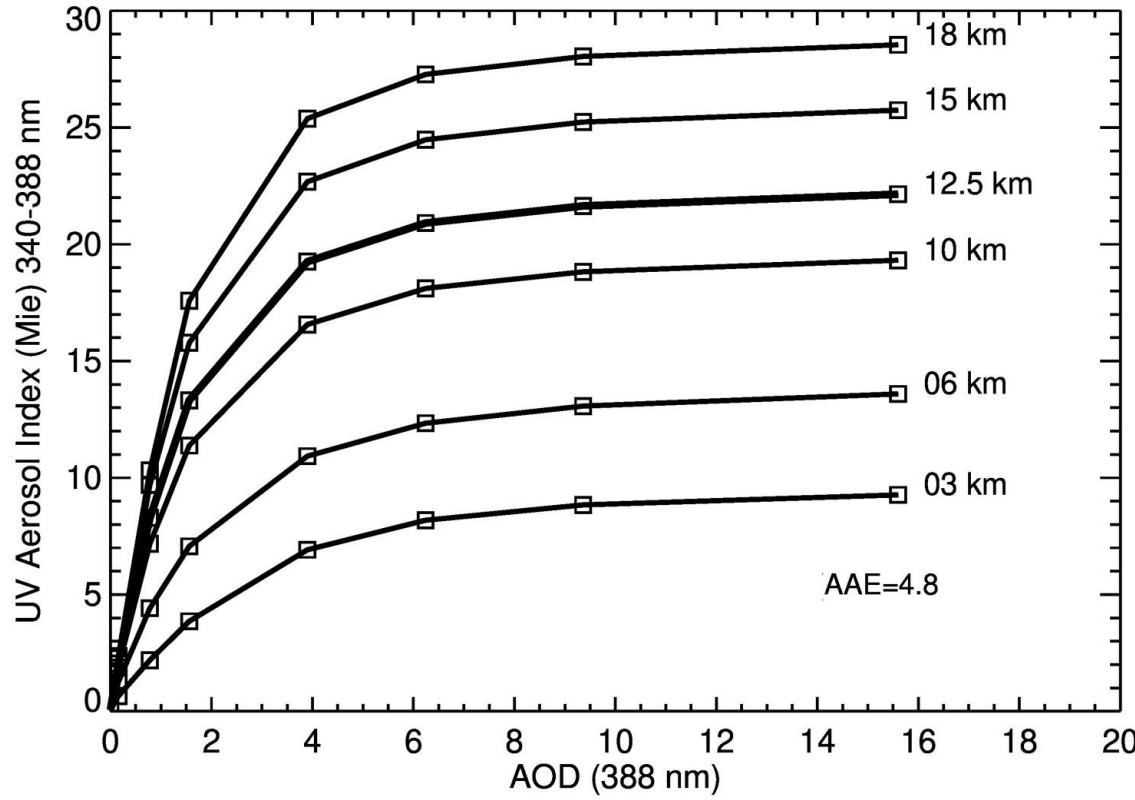

**Figure 1. Modelled relationship between UVAI and AOD as a function of ALH for carbonaceous aerosols of**
**assumed 340-388 nm aerosol absorption exponent of 4.8 (see text for details).**

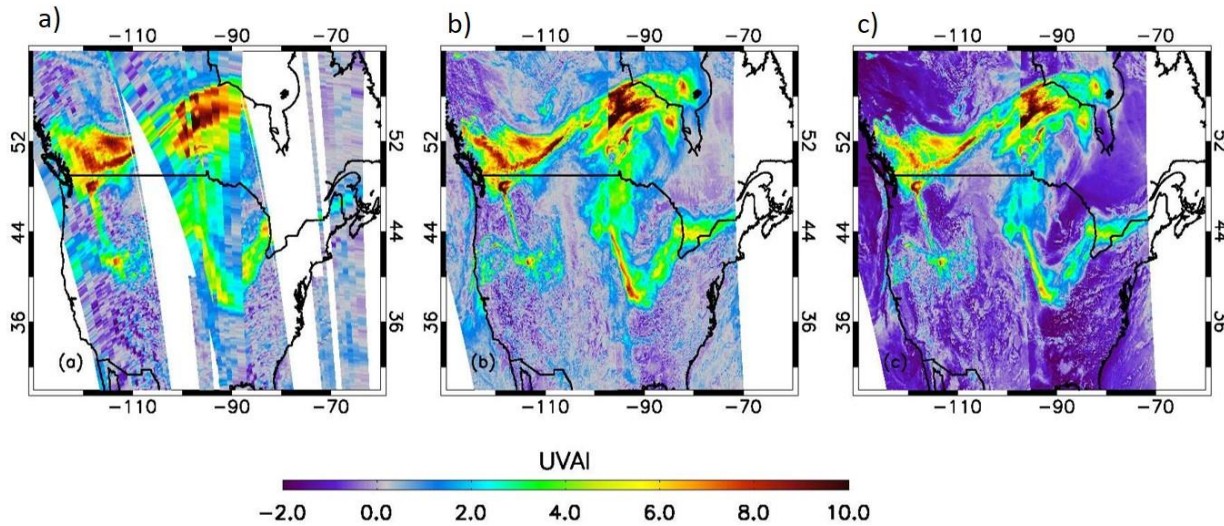

Figure 2. Observed UVAI on August 18, 2018 over North America from a) OMI observations, b) TROPOMI observations using the NASA algorithm and, c) TROPOMI operational ESA/KNMI product.

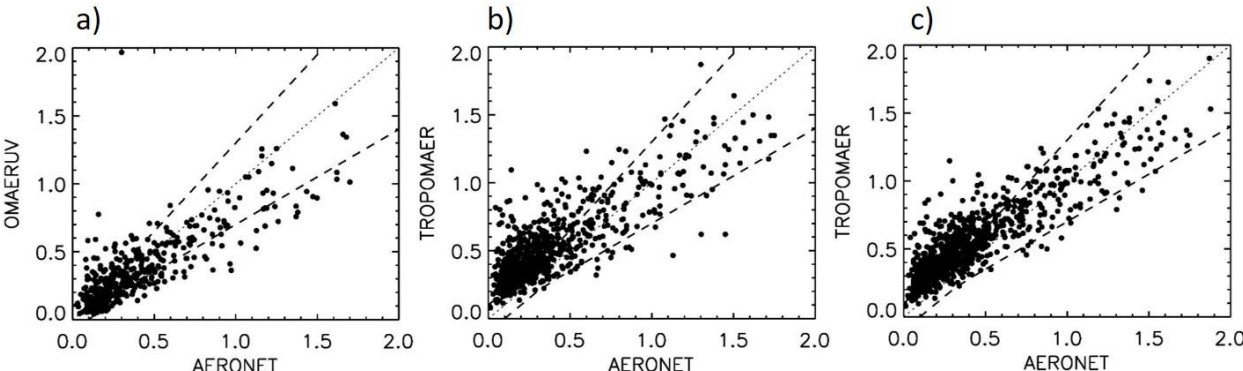

**Figure 3.** AERONET – satellite comparisons of OMI retrieved 388 nm AOD (a), TROPOMI using heritage cloud screening (b), and TROPOMI using VIIRS cloud mask (c). Dotted line indicates the one-to-one line, and dashed lines represent expected retrieval uncertainty (largest of 0.1 or 30%). See text and Table 2 for details.

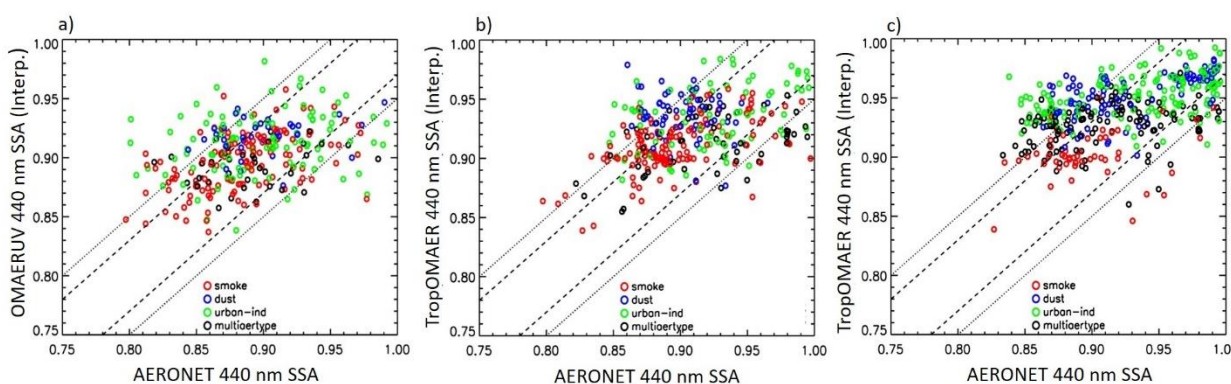

Figure 4. As in Figure 3 for single scattering albedo of dust aerosols (blue), smoke aerosols (red), urban-industrial aerosols (green), and aerosol mixtures (black). Dashed line indicates agreement between ±0.03, solid line indicates agreement between ±0.05.

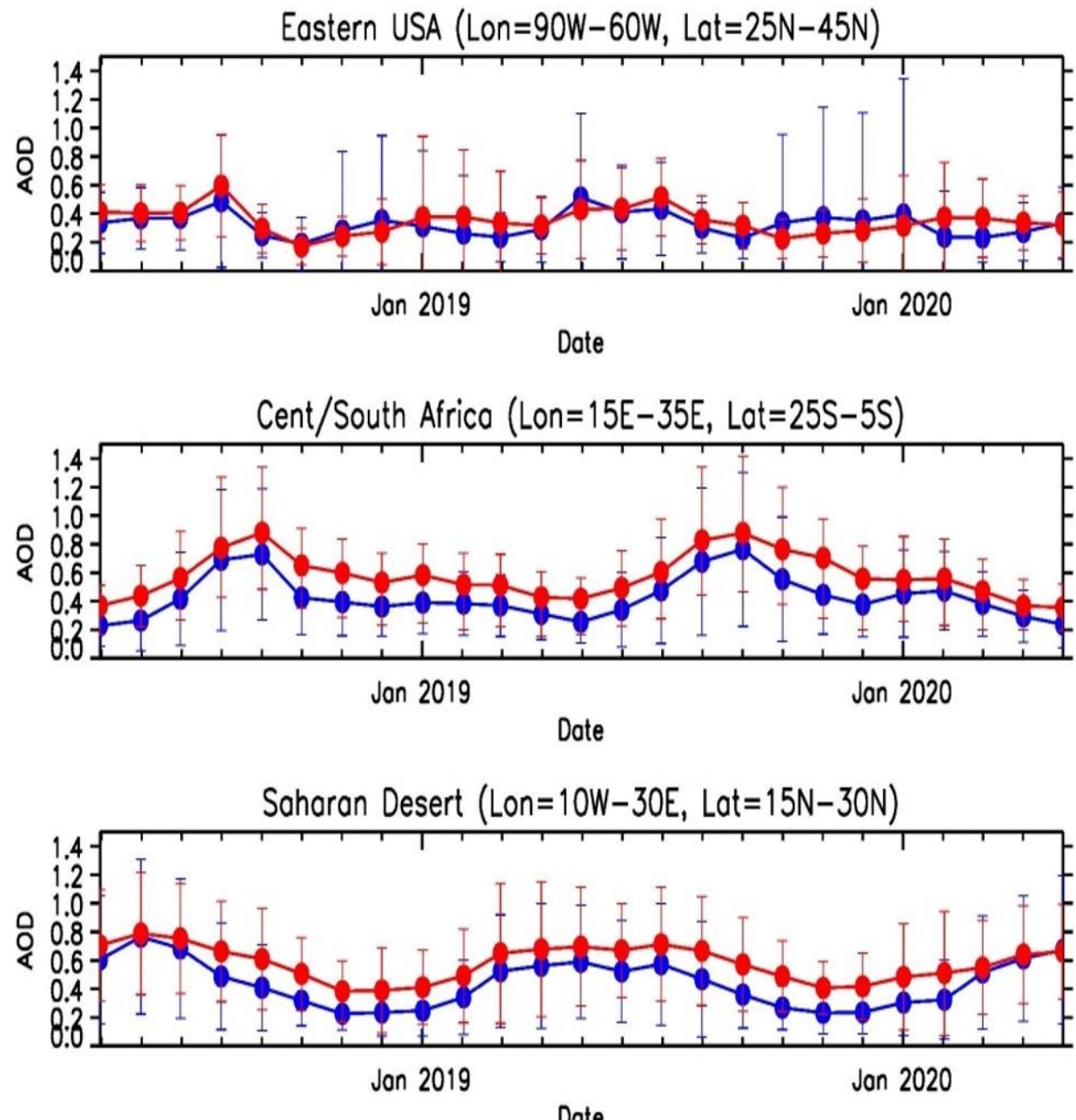

**Figure 5.** Two-year time series of monthly average OMI (in red) and TROPOMI (in blue) 388 nm AOD values for Eastern United States (top), Southern Africa (middle), and Saharan Desert (bottom). Vertical lines indicate standard deviation of the mean associated with both temporal and spatial variability.

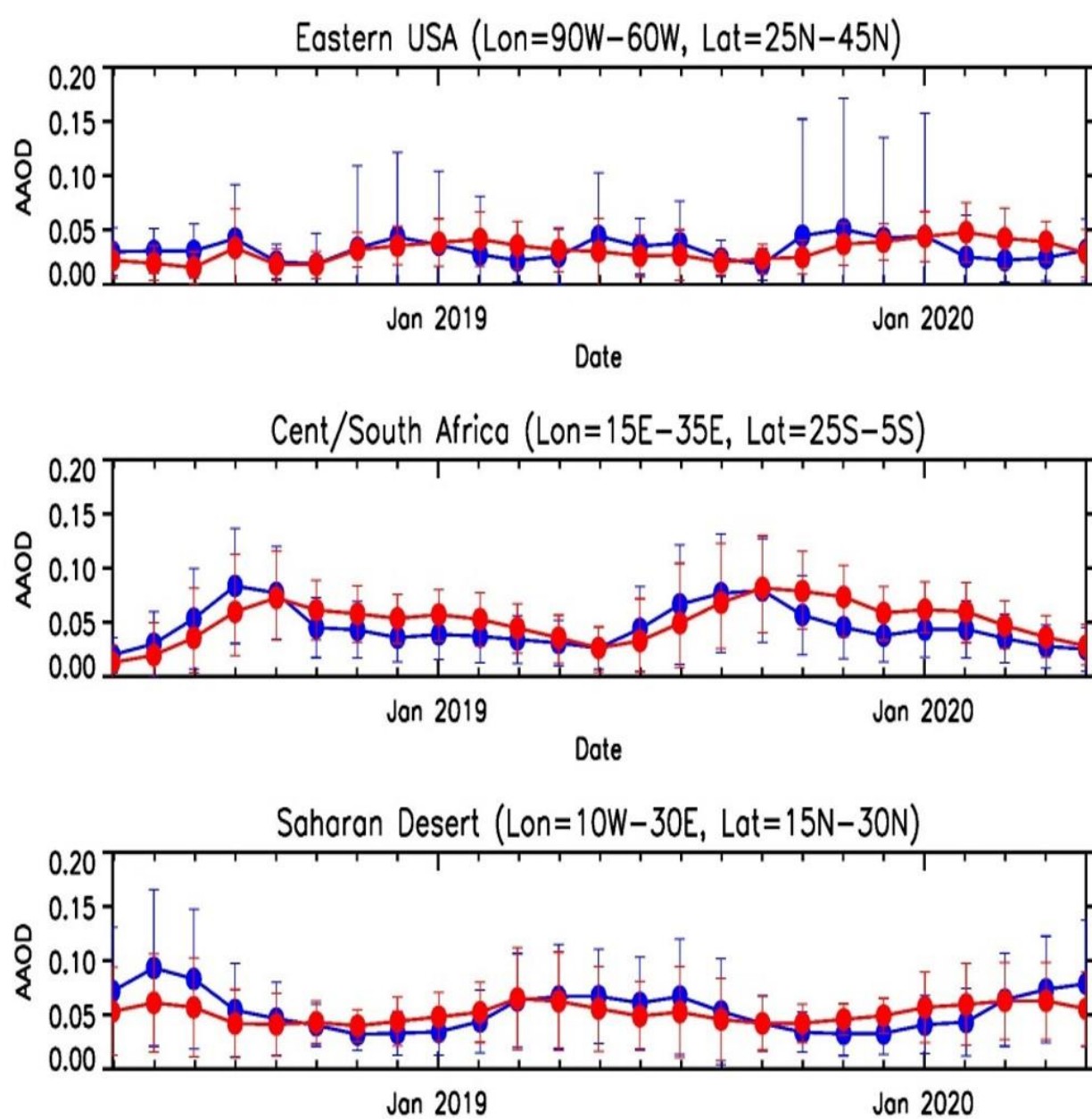

2  **Figure 6.** As in Figure 5 for 388 nm AAOD.

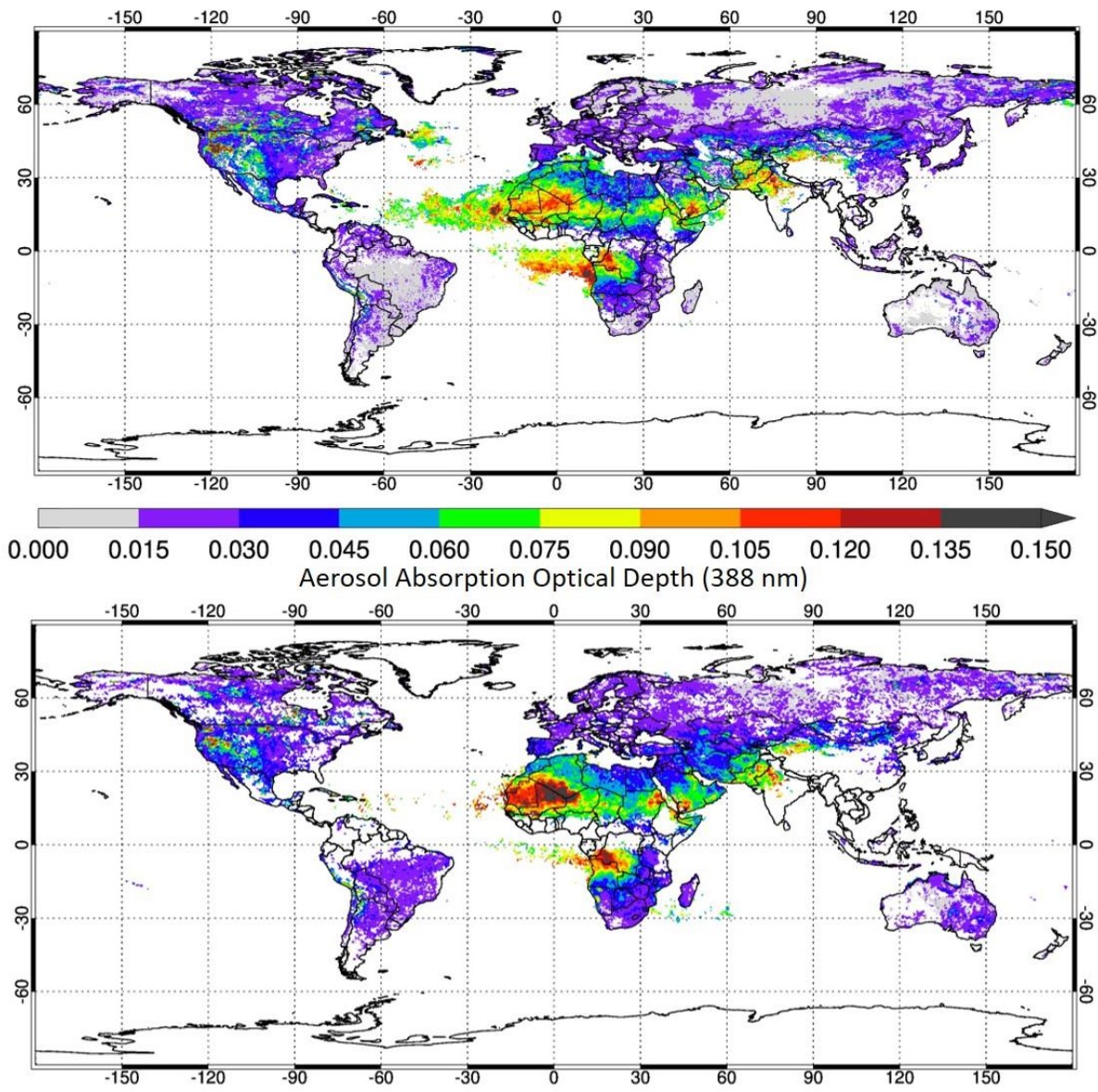

2 **Figure 7**. NH Summer Season (June-July-August 2018) global map of 388 nm Aerosol Absorption
3 Optical Depth from TROPOMI (top) and OMI (bottom) observations.

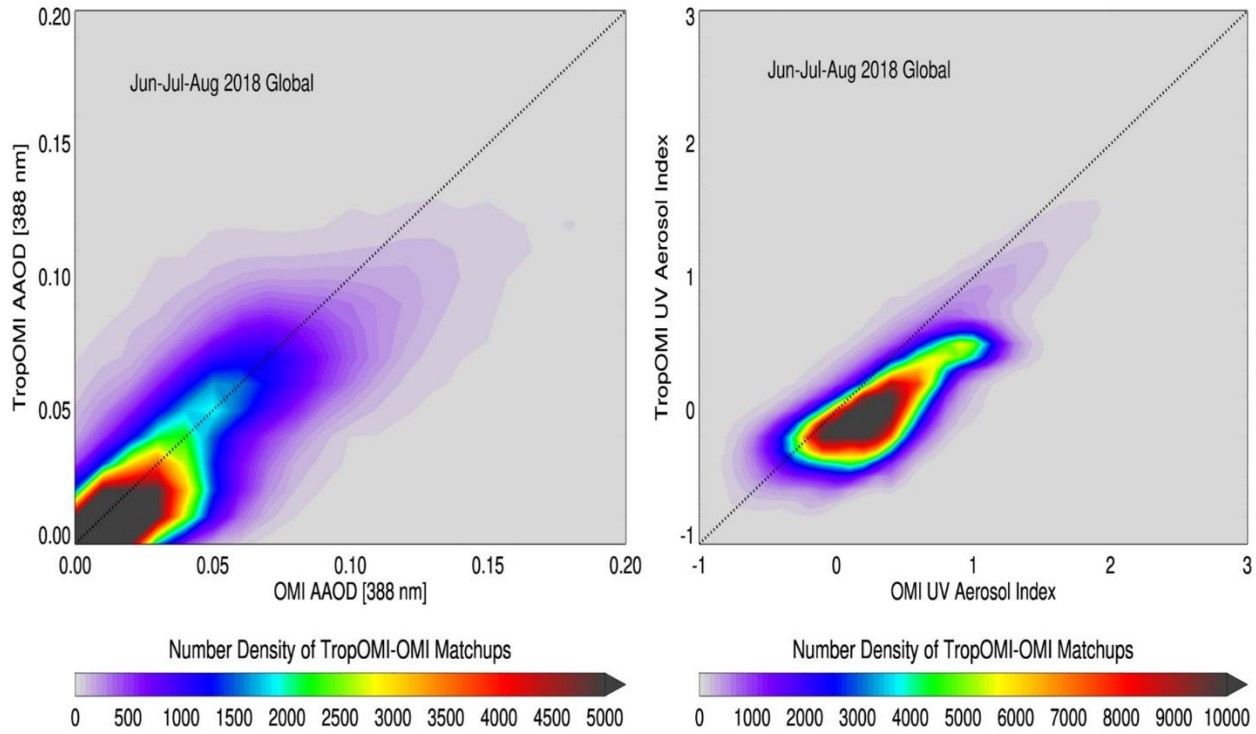

Figure 8. Density plots of OMI (x-axis) and TROPOMI (y-axis) gridded monthly mean (June, July, August 2018) values of 388 nm AAOD (left) and UVAI (right). Dotted line indicates one-to-one line of agreement.

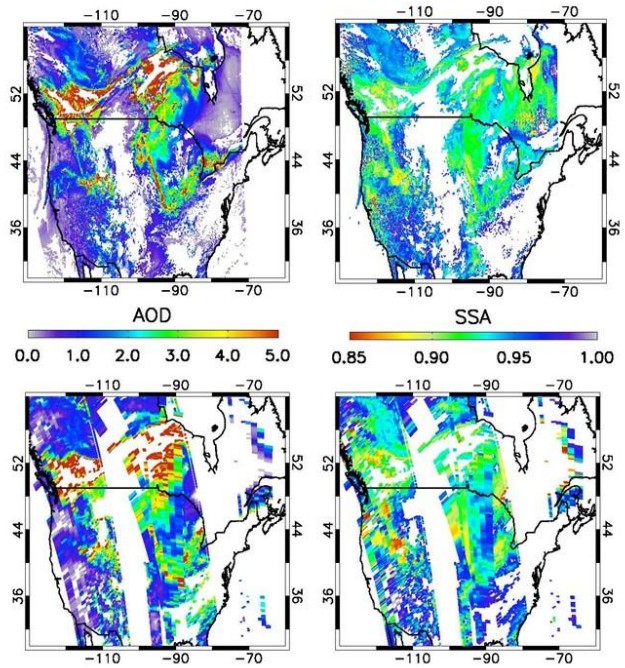

3    **Figure 9.** Spatial Distribution of 388 nm AOD (left) and SSA (right) on August 18, 2018 derived from
4    TROPOMI (top) and OMI (bottom) observations.

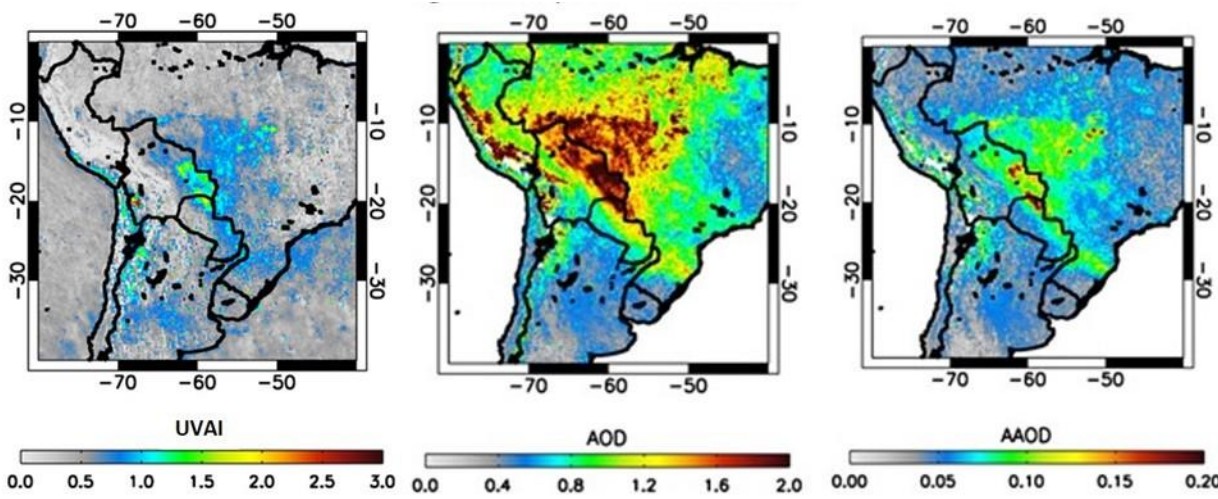

2 **Figure 10.** September 2019 monthly average values of TROPOMI UVAI (left), 388 nm AOD (center)
3 and AAOD (right) over South America.

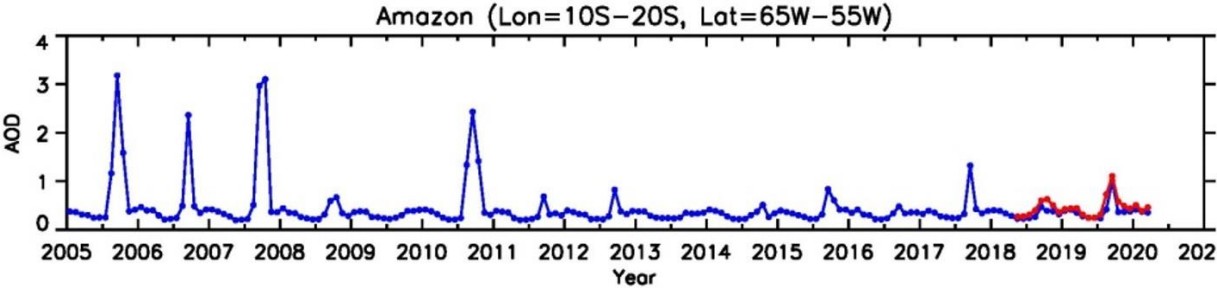

2  **Figure 11.** Time series of 388 nm AOD over the amazon basin from OMI (blue line) and TROPOMI (red line)

3  observations.

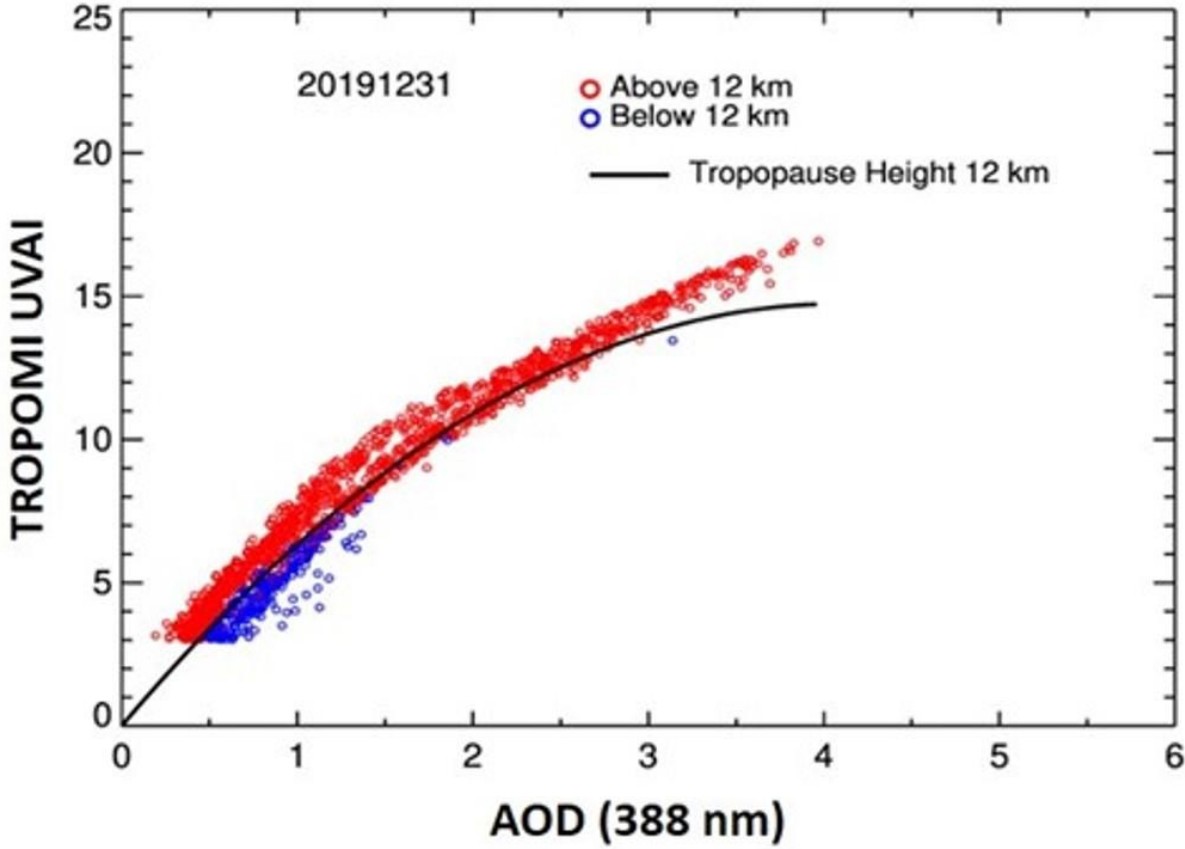

3   **Figure 12.** UVAI-AOD relationship at ALH 12 km for the 2019-2020 Australian fires (black line)  on December 31,

4   2019. Red symbols represent aerosol retrievals at 12 km and higher. Blue symbols indicate retrievals at heights

5   lower than 12 km.

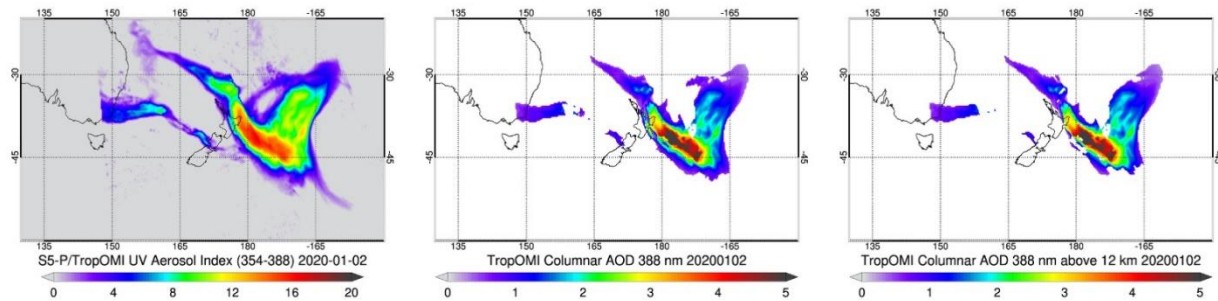

3   **Figure 13.** TROPOMI UVAI (left), total column 388 nm AOD (center) and above 12 km AOD (right) fields of
4   Australian smoke plume on January 2, 2020.

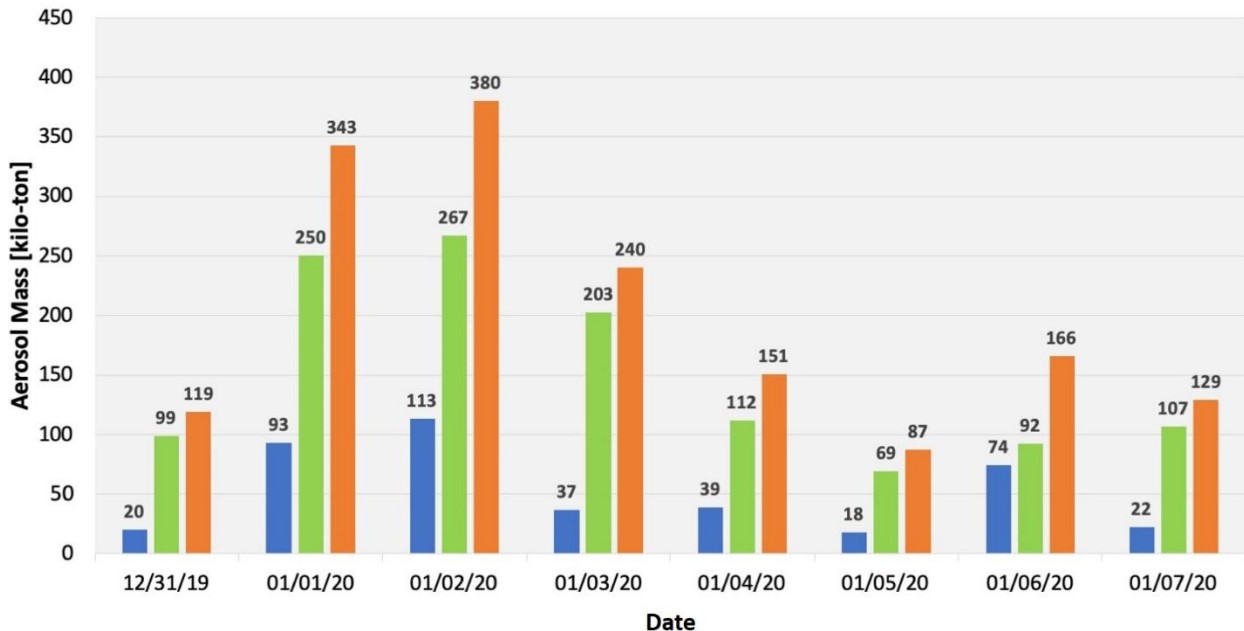

2 **Figure 14.** Calculated Daily aerosol mass (kilotons) in the stratosphere from TROPOMI observations, from

3 December 31, 2019 to January 7, 2020. Results are reported for aerosols in cloud-free conditions (blue bars),

4 aerosol above cloudy scenes (green bars), and their sum (orange bars).

**Appendix A**

2         **AERONET-TROPOMI Comparisons at individual sites**

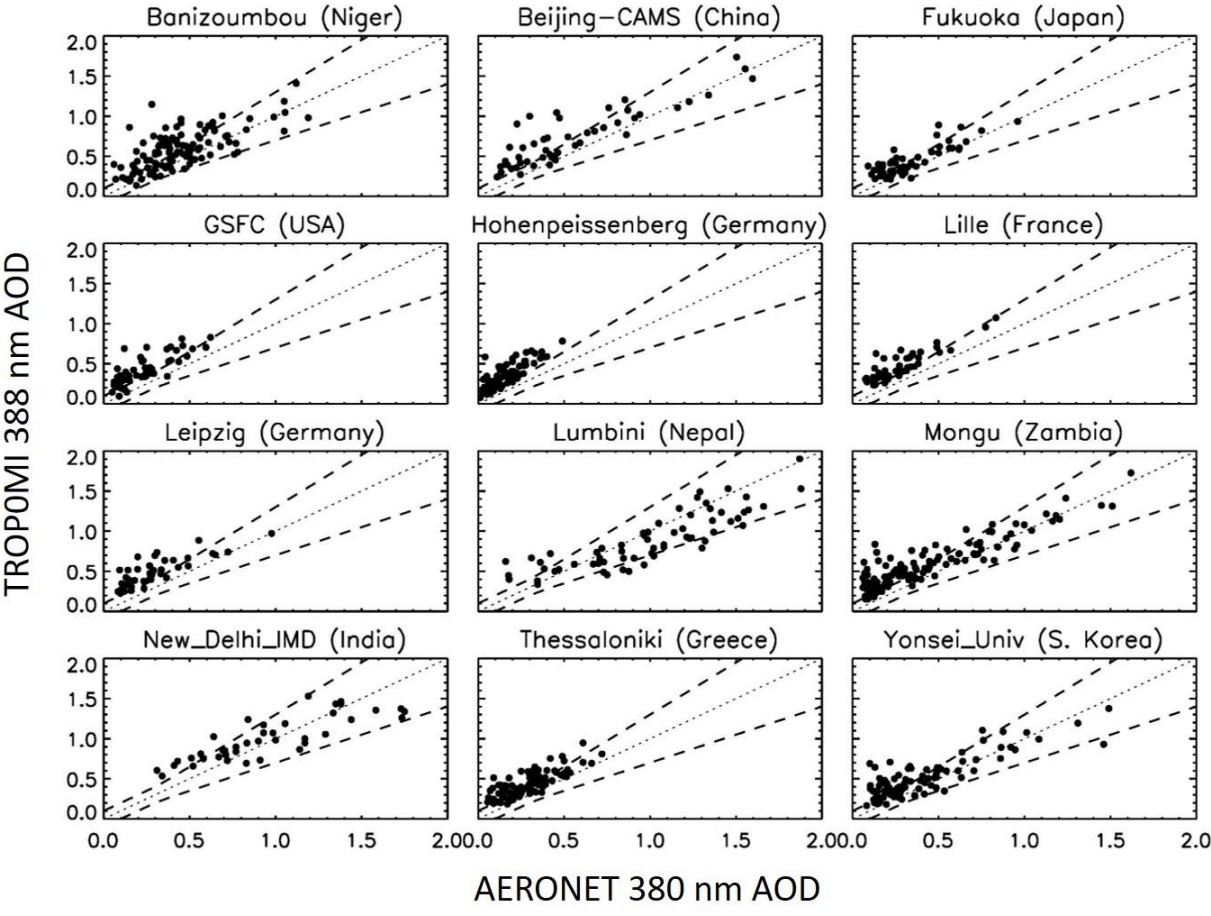

**Figure A.1.** Scatter plots of AERONET measured 380 nm AOD (x-axis) and TROPOMI retrieved 388 nm AOD (y-
axis) at each of the sites used in the analysis. Dotted line indicates the one-to-one line, and dashed  lines represent
expected retrieval uncertainty (largest of 0.1 or 30%).

                                              **Appendix B**

**Extinction to mass conversion**

The total aerosol mass injected in the stratosphere, M, can be estimated by converting stratospheric AOD ($\tau_{str}$, see
below) into an equivalent aerosol mass per unit area, using the equation (Krotkov et al.,  1999)

$$M = \Sigma \frac{4}{3} \rho r_{eff} A \tau_{str} f(\lambda, r_{eff}) \qquad \text{(B-1)}$$

that yields the summation of the aerosol mass over the total area covered by the aerosol plume. In Equation B-1, $\rho$ is
the aerosol particle mass density in g-cm$^{-3}$, $r_{eff}$ is the effective radius ($\mu$m) associated with the particle size
distribution (van de Hulst, 1957), $A$  is the effective  geographical area in km$^2$ associated with the retrieved
stratospheric AOD averaged over each 0.25ºx0.25º lat.-lon. grid (see text for details), and $f(\lambda, r_{eff})$ is a dimensionless
extinction-to-mass conversion factor, averaging over particle size distribution, defined as

$$f = \int_0^\infty r^2 n(r) \partial r / \int_0^\infty r^2 Q_{ext}(\lambda, r) n(r) \partial r \quad \text{(B-2)}$$

14    where $n(r)dr$ is the assumed number particle size distribution and $Q_{ext}(\lambda, r)$ is the extinction efficiency factor
15    calculated using Mie theory. Calculations were carried out for particle mass density values of 0.79 and 1.53 g-cm$^{-3}$
16    which cover the range of values reported in the literature (Reid et al., 2005).

