# Peer review of "TROPOMI Aerosol Products: Evaluation and Observations of"

_Atmospheric Measurement Techniques, 2020_

## Referee Comment (RC1) · Anonymous Referee #1 · 3 Jun 2020

Summary: This manuscript introduces the TropOMAER aerosol retrieval algorithm. The algorithm is essentially the heritage OMAERUV algorithm from the OMI collection, now modified to be applied to TropOMI data instead. In this adaptation process, the ability to retrieve above cloud aerosol OMACA has been included. The introduction to the algorithm itself is quick. The authors point out two major differences from OMAERUV: (1) TropOMI's finer spatial resolution (2) still evolving radiometric calibration. There is a quick evaluation section showing TropOMAER retrievals against 12 selected individual AERONET stations for aerosol optical depth (AOD) and an aggregation of all 12 stations for single scattering albedo (SSA). Then the bulk of the manuscript demonstrates TropOMAER in three interesting and newsworthy biomass

burning events.

Assessment: There is much merit in this manuscript. The three examples, especially the third example, are scientifically extremely interesting. However as currently written, it is missing too much detail for publication in AMT. AMT is where algorithm developers, such as these authors and myself "talk shop", and where we document the details of algorithms and validity of our products. While the heritage algorithms are well-documented in the literature, porting an algorithm to a new sensor introduces new challenges that are very interesting to other algorithm developers and should be included in a paper like this one. This manuscript could easily be adapted into a form that would be appropriate for AMT, if that is what the authors want to do.

These are the points that would make the manuscript ready for publication in AMT:

(1) much more description of the algorithm itself, even if that description were partly reiterated from previous publications.

(2) highlight differences between OMI and TropOMI instruments, between OMIAERUV and TropOMAER algorithms, most importantly between results from each sensor. Of prime interest to potential users of TropOMAER products who have been using OMI products is how do the products from the new sensor compare with the products from the old sensor. The only place I see a hint of that is the plotting of OMI retrievals with TropOMI retrievals on the time series in Fig. 5. However, that figure is not satisfying. Much more interesting than the 15-year time series would be a difference time series during the TropOMI era and a scatter plot of TropOMI against OMI, even on a monthly mean basis.

(3) evaluation of TropOMAER should be expanded. There should be an effort to trace the consequences of the finer spatial resolution and issues with calibration to the evaluation. Right now the authors skirt these issues without really proving anything.

For example they mention subpixel cloud contamination being absent in most of the
validation sites. However, when I look at the 12 panels in Figure 1, I see no qualitative difference between the 3 sites mentioned as having subpixel cloud contamination and the other 9 sites. If there was marked improvement from Ahn et al., 2014, then that improvement should be demonstrated in this paper. I should not have to call up that paper and run my eyes between two different figures in two different papers to see the improvement.

Later they mention needing a finer resolution surface albedo map, and there is also mention of the calibration causing some of the offset in the validation plots. Each of these issues is very interesting to another algorithm developer, like myself, or to potential users of the products. AMT is the right journal to present an analysis of these issues, and prove their consequence on the retrievals. Currently that analysis is missing.

(4) Slow down and present the details. I felt that there was a rush through the "boring" algorithm piece of the paper in order to get to the "exciting" demonstration with the big biomass burning events. There are many details left behind in the rush:

There are many acronyms never properly introduced:

p.2 line 2 should put (SWIR) after shortwave infrared. P2 line 5. ESA and DLR? P2 line 28. Should put (ALH) after aerosol layer height P5 line 5. UVAI is never defined as an acronym, and worse, it is never defined as a product. Suddenly it is being shown in figures and being used as a fundamental part of the analysis. P6 line 25 SAM? P6 line 33. What are total mappers?

The concepts of Level 1 and Level 2 data are not explained (p2 line 5). Exactly what AERONET data are we looking at? Version 2 or 3? Levels 1.5 or 2? There is no explanation that AERONET AOD has a documented uncertainty of 0.02 in the UV, but that the SSA retrieval is a retrieval with much broader error bars. There is no explanation of why or how these 12 stations are selected, nor what the time range we are looking at.

(5) Provide more detail in the demonstration section.

Figure 3 would benefit greatly by adding a swath just to the west of the swath shown. Right now there is a lot of description of fires and smoke in California, the Pacific Northwest and British Columbia, but none of those areas are shown in the figure. Only the areas downwind.

P6. Lines 1 to 6. Is this method here the manifestation of the ACA part of the TropOMI retrieval that is mentioned at the beginning? If so, then please make that clear. If it is a different method, then explain why the referenced ACA method is not used. If not, then is there any demonstration of the ACA TropOMI method? ACA is an important new addition to OMIAERUV, and should be highlighted or discussed if this is going to AMT.

P6 Line 10. The extinction-to-mass conversion is important. The appendix should be referenced here.

P6 lines 13-16. Is there a physical basis for this? This is important, and how the UVAI-AOD relationship relates to height, and especially to height in the stratosphere needs to be explained. Remember that UVAI jumps in suddenly with no introduction. It would be worthwhile to take the time to explain it, and some of the physics behind the whole interrelationship between height, AOD, UVAI and absorption. Maybe in Section 2?

P6 line 25 to P7 line 2. A lot of numbers are given here and these are means with uncertainties surrounding them. The uncertainty is given at the end of $\pm 40\%$. It would be helpful to explain how the mean is derived (for what density) and what is the interplay between assumptions of density and uncertainty in height.

P7 lines 27-33. This is very interesting, but the figure doesn't really portray this information well. Figure 5 needs to become more informative.

(6) All the captions need to more descriptive. Be sure to give details on specific data, be sure to describe what is shown in each panel, what wavelength is being shown,

what temporal resolution is being plotted (fig. 5), what do each of the colors in the color bars represent. But in general a LOT more information needs to be in the figure captions.

Suggestion: It occurred to me that this manuscript might fit a "letters" journal much better. Right now it is not too long. The authors would need to triage their figures down to 4. Perhaps Figs. 1, 3, 5 (with a bottom panel showing the difference between TropOMI and OMI) and 8. Then the very short description of the algorithm, evaluation and methods would be appropriate, and the purpose of the paper is NOT to describe TropOMAER, but to illustrate these biomass burning events. The point of the paper shifts from an "atmospheric measurement technique" to a better understanding of the Earth's atmospheric phenomena. GRL would be a possibility, but also ERL.

---

## Referee Comment (RC2) · Anonymous Referee #2 · 10 Jun 2020

This paper briefly introduces a TropOMI aerosol data set based on heritage OMI UV algorithms by the Torres group (OMAERUV and OMACA). This provides UV aerosol index (UVAI), aerosol optical depth (AOD), and single scattering albedo (SSA). A comparison of AOD and SSA against data from selected AERONET sites is presented, along with a few case studies of extreme events. The concept of the paper is in scope for AMT. The quality of language is good. The topic is important because OMI is ageing and TropOMI is the next generation of this type of sensor (OMPS on SNPP and JPSS has some aerosol capabilities but is in other ways worse than OMI).

However, honestly, the current paper feels more like a conference proceedings or an

article for a Letters journal than a full scientific paper. It is brief and does not go into much detail. For a focused journal like AMT I think something much more technical is needed. Though I realise I am proposing a fair amount of work, I prefer that the authors expand this analysis rather than resubmit elsewhere, because I think a thorough accounting for TropOMI's capabilities for UV aerosol remote sensing is needed and is more or less missing from the literature. The authors are the right people to do this because they are the most expert with their data products. I know it is annoying when reviewers ask to do more work, but there is not enough content here to justify publication and I don't think that the article as written satisfies the scope a reader would reasonably expect. Case studies are one thing but by nature are typically unusual events and so looking at them may not give a representative picture of the data set as a whole. I recommend major revisions and would like to review the revision.

My main suggestion for expansion is to give a detailed comparison between OMI and TropOMI results. Users familiar with OMI need to know whether we can use TropOMI for the same types of research, and to what extent the same caveats/biases are found. Right now this is not answered in a thorough way. One big advantage of TropOMI over OMI is the spatial resolution. I would expect that this is important because those cases where the UV technique works well (absorbing aerosols) are also often strong and heterogeneous events. So the finer spatial resolution might mean both (1) less cloud contamination and (2) better AOD/SSA retrievals, because top of atmosphere radiance is not linear in AOD, so by resolving more spatial structure you become less sensitive to sub-pixel variations. If this is true in practice, great. If not, this needs to be shown and understood. It is briefly discussed in Section 3.1 but not supported by the plots shown, only by briefly mentioning other references. Here are some suggestions for relevant analyses to include:

(1) Show global maps so we can see how similar the big picture looks from both sensors. In my view the time series in Figure 5 isn't sufficient here because both data sets are heavily spatially and temporally averaged in it.

[Figure]

(2) Include OMI in some of the case studies (e.g. visual inspection of maps).

(3) OMI validation results could be presented alongside the TropOMI data. I know the validation has been published elsewhere but it will be clearer to the reader if plots are shown next to one another with the same axis range, etc.

(4) Directly plot (as a scatter density diagram) the AOD and/or UVAI from OMI and TropOMI, for collocated pixels (i.e. same scene, same time, similar geometry) at level 2 resolution. The orbits should overlap frequently. Then we can see if there's much scatter, if it's a straight line or not, etc. I don't know how much collocated data is needed to get a meaningful comparison – perhaps the case studies give enough, perhaps it has to be done on a month's worth of data. MODIS or VIIRS data could be useful for extra context (and filtering); I know and the manuscript mentions that the TropOMI orbit choice makes it possible to take advantage of SNPP VIIRS for e.g. cloud masking.

The above comments and suggestions all apply (potentially) to the DSCOVR-EPIC sensor, too, although OMI is the more well-known and mature record so probably makes better sense to baseline against. Though I would certainly be happy to see a three-way (OMI, EPIC, TropOMI) comparison!

Other comments on the study are as follows:

Introduction or section 2.1: somewhere here it would be good to contrast TropOMI capabilities (e.g. spatial/spatial) with OMI and maybe TOMS and EPIC, since those are the main comparative products. The introduction mentions GOME and SCIAMACHY but those are less relevant since the authors' algorithms are from TOMS/OMI heritage and EPIC data are shown later. Maybe mention OMPS too as while a step backwards from OMI in terms of spatial resolution, it is used for UVAI and is the US operational follow-on for that. I know that there are TropOMI products in development on the Dutch side too – it's not clear to me whether those are public yet, but if so, there may be value in comparing and contrasting with those too.

Section 2.2: if I understand correctly this section states that (1) there is a 5-10% calibration difference between OMI/OMPS and TropOMI in the relevant bands in the standard calibration, and (2) because of this the authors do their own vicarious calibration. Is that right? Either way, this could be worded a little more clearly. What is the difference between the sensors after the vicarious calibration?

Section 3: clear statements and references about AERONET data products and versions used need to be made. For example, I assume this is version 3 level 2.0 direct Sun (Giles et al AMT 2019) and inversions (Sinyuk et al AMT 2020). However this does not appear to be actually stated in the paper. If this was not the versions used, the analyses should be repeated using the latest data versions.

Section 3.1: if the authors really believe that a relative uncertainty of 30% on TropOMI AOD is true, then by definition they should not be using linear least squares regression fits, because a relative uncertainty means that the assumption of constant variance of errors is broken. See for example standard statistics textbooks or web pages such as https://statisticsbyjim.com/regression/heteroscedasticity-regression/ . This issue could be addressed with weighted least squares. Ideally also the uncertainty on AERONET AOD (I think 0.02 in this spectral region) should be accounted for in the fitting. Also, if you expect a relative uncertainty then RMSE is not the best metric to be reporting since that is scale-dependent. . . others like relative RMSE would be more appropriate to quote instead/as well (and this would help tell you if it is really 30%). The statistical analysis here is not very appropriate. The authors may have used this type of analysis before but that does not mean it is ok to do something again if it is wrong.

Section 3.2: the authors use a 6 hour time window (3 hours each side) for the SSA comparison because morning/evening almucantar inversions have lower uncertainty than midday ones. The untested assumption here is that SSA does not vary much throughout the day. Ok, but version 3 also introduced hybrid scans which were specifically developed to solve this problem by sampling a larger air mass and scattering angle range during the middle of the day. This could be checked by using the hybrid inversions as well and seeing if you get the same results. Also, an explanation is needed for how the authors split the data into the three aerosol type categories for Figure 2 and the discussion.

Section 4: this feels like advertising. I agree that TropOMI results look impressive but (aside from a brief mention of AERONET AOD) there is no way to know how 'real' they are. This section feels like something you might put on a webpage or brochure to attract attention to your new data set, rather than a detailed scientific analysis. I am not sure what is best to do here. For a journal like AMT I'd rather than space was devoted to more technical, large-scale comparisons. Perhaps this aspect could be split off for a Letters journal. Or, expanded with more context from meteorology and other (space or suborbital) data records and submitted separately to ACP. I know this is a joint special issue but the content still needs to match the journal. It does not really fit here, and there's not enough detail presented to consider this paper an authoritative reference for these case studies.

Figure 6 and associated text: I'm not sure that it makes sense to show the EPIC results on the left panel. That's a different sensor, different resolution, different observation geometry (backscatter for EPIC). UVAI is sensitive to all of these things. Also, what is the scaling referred to in the left panel? That is not mentioned in the paper. I expect that the general point about the two events will still stand but it's not clear how much of the systematic difference (and scatter on the left panel) are a function of real differences in the smoke in the two events and how much is contributed by sensor differences. The paper is far too sparse in detail for a reader to judge, which makes the comparison less instructive.

Figure 6 legend: is the black dot in the left panel legend (12 km) meant to be a black line like in the right panel? If so, formatting should be consistent. If not, the difference needs to be explained.

Section 5: "The NASA TropOMAER aerosol algorithm is a modified version of the one

applied to OMI observations." Wait, what? Section 2 describes the OMI approach but doesn't clearly state that there are modifications. What are these modifications, why were they made, what effect does this have on the results, and will they be back-ported to OMI? This all needs to be addressed in the paper.

—————————————————————

---

## Referee Comment (RC3) · Anonymous Referee #3 · 15 Jun 2020

This paper presents NASA aerosol product for TROPOMI obtained with TropOMAER retrieval algorithm.

In general, the manuscript is well-written, well-structured and demonstrates the possibilities of TropOMAER retrieval algorithm. First, the AOD and SSA products were evaluated using AERONET dataset for 12 representative sites. Then, the results of the algorithm application to a few important aerosol events were presented and total aerosol mass injection was estimated.

There are few remarks regarding AOD and SSA validation against AERONET.

1. Figure 1 and Table 1 clearly indicate the presence of positive bias in TropOMAER

[Figure]

AOD product at 380 nm over all 12 representative sites. Authors already provided some guess about the origin of this bias and mention that this issue is under investigations. Nevertheless, since the retrieval is carried out at 388 nm, and reported also at 354 and 500 nm, presenting AOD validation results in the manuscript for two wavelengths (for example, 380 and 500 nm) would be very useful to address the bias issue.

2. One of the parameters of AOD evaluation is 30% matchup criteria. What is the origin of these criteria? Is AOD product with 30% uncertainty sufficient for trace gases retrieval? For example, GCOS requirements on AOD are much more strict: 0.03 or 10%.

3. The results of SSA validation show reasonable correspondence with AERONET. Nevertheless, Figure 2 clearly shows overestimation of SSA especially for absorbing aerosol when SSA from AERONET < 0.9. Is this related to the same issues providing positive bias in AOD? Is this SSA overestimation a demonstration of limitation of aerosol model used in TropOMAER algorithm? More discussions here are necessary.

In general, I would recommend authors to reserve some space in the manuscript for discussions regarding identified issues in the retrieval. For example, the mentioned above issues for AOD and SSA retrieval as well as authors thoughts how to treat these issues would be highly appreciated by broad remote sensing community. These discussions would greatly increase the scientific strength of the paper.
* * *

---

## Author Comment (AC1) · 16 Aug 2020

As a result of the review process, the manuscript has been modified significantly. Mayor changes are:

1) Section 2 of the paper has been extended to include a brief but detailed description of the TropOMAER algorithm. It includes a description of the UVAI calculation as well as a summary of the AOD/SSA retrieval process.
2) Section 3 on the validation of retrieval results using AERONET observations also changed considerably. The original validation analysis consisting of a direct validation of TROPOMI AOD results to AERONET observations at 12 sites was replaced with an approach that allows the separate evaluation of retrieved product improvement as a result of instrument enhancement and algorithmic improvement. AERONET observations 12 sites are used as an aggregate. A three way validation exercise is then carried out: 1) AERONET vs OMI, 2) AERONET vs TROPOMI using heritage (OMI) cloud mask, and 3) AERONET vs TROPOMI using VIIRS-based cloud mask. Inter-comparison for validations 1 and 2 highlights the effect of improved instrumental capabilities, whereas differences in validations 2 and 3 indicate retrieved product improvement due to algorithmic upgrades.
3) The revised paper (to be available soon after the submission of replies to reviewers' comments) contains 13 figures (five more than in the original version).

In the reply below the reviewer's comment is in black and our answer in blue.

Reply to Comments by Reviewer 1

Summary:
This manuscript introduces the TropOMAER aerosol retrieval algorithm. The algorithm is essentially the heritage OMAERUV algorithm from the OMI collection, now modified to be applied to TropOMI data instead. In this adaptation process, the ability to retrieve above cloud aerosol OMACA has been included. The introduction to the algorithm itself is quick. The authors point out two major differences from OMAERUV: (1) TropOMI's finer spatial resolution (2) still evolving radiometric calibration. There is a quick evaluation section showing TropOMAER retrievals against 12 selected individual AERONET stations for aerosol optical depth (AOD) and an aggregation of all 12 stations for single scattering albedo (SSA). Then the bulk of the manuscript demonstrates TropOMAER in three interesting and newsworthy biomass burning events.

We thank the reviewer for his/her comments that have contributed to an improved manuscript.

Assessment:
There is much merit in this manuscript. The three examples, especially the third example, are scientifically extremely interesting. However as currently written, it is missing too much detail for publication in AMT. AMT is where algorithm developers, such as these authors and myself "talk shop", and where we document the details of algorithms and validity of our products. While the heritage algorithms are well-documented in the literature, porting an algorithm to a new sensor introduces new challenges that are very interesting to other algorithm developers and should be included in a paper like this one. This manuscript could easily be adapted into a form that would be appropriate for AMT, if that is what the authors want to do. These are the points that would make the manuscript ready for publication in AMT:

(1) much more description of the algorithm itself, even if that description were partly reiterated from previous publications.

The section on algorithm description was extended to elaborate on key aspects of the inversion scheme.

(2)  highlight differences between OMI and TropOMI instruments, between OMIAERUV and TropOMAER algorithms, most importantly between results from each sensor.

The purpose of the comparison to AERONET has changed from  the narrowly focused AOD validation exercise in the original version of the paper, to an analysis of  the instrumental and algorithmic differences throughout the use of independent ground-based observations. The combined AERONET data aggregate from observations the 12 sites, is compared to satellite observations as follows. An evaluation of instrument-related and algorithmic improvements is done by comparing AERONET measurements to three satellite-based data sets:1) OMAERUV, 2) TropOMAER with heritage (i.e., OMAERUV) cloud screening, and 3)  TropOMAER with VIIRS cloud mask.

A comparative analysis of evaluations 1 and 2 shows the impact of enhanced instrumental capabilities,  whereas the analysis of evaluations  2 and 3 highlights the effect of using the VIIRS cloud mask which is the only TropOMAER algorithmic modification.

Of prime interest to potential users of TropOMAER products who have been using OMI products is how do the products from the new sensor compare with the products from the old sensor. The only place I see a hint of that is the plotting of OMI retrievals with TropOMI retrievals on the time series in Fig. 5. However, that figure is not satisfying.  Much more interesting than the 15-year time series would be a difference time series during the TropOMI era and a scatter plot of TropOMI against OMI, even on a monthly mean basis.

The parallel validation of OMI and TROPOMI described above addresses this issue.

As suggested, the consistency of the OMAERUV and TropOMAER records are evaluated by comparisons between the products at different time scales:

OMI-TROPOMI visual inspection comparisons of UVAI are shown on Figure 1 for the smoke plume over North America on August 18, 2018. This comparison also includes the KNMI TROPOMI UVAI.

Side-by-side maps of OMI and TROPOMI retrieved SSA and AOD for the same event are also shown on Figure 8.

A two-year time series of monthly-averaged OMI and TROPOMI AOD and AAOD  over three regions are shown on Figure 4.

OMI and TROPOMI summer seasonal global maps are compared in Fig 6, and a scatter plot of OMI-TROPOMI monthly UVAI values is shown on Figure 7.

(3)  evaluation of TropOMAER should be expanded. There should be an effort to trace the consequences of the finer spatial resolution and issues with calibration to the evaluation. Right now the authors skirt these issues without really proving anything. For example they mention subpixel cloud contamination being absent in most validation sites. However, when I look at the 12 panels in Figure 1, I see no qualitative difference between the 3 sites mentioned as having subpixel cloud contamination and the other 9 sites. If there was marked improvement from Ahn

et al., 2014, then that improvement should be demonstrated in this paper. I should not have to call up that paper and run my eyes between two different figures in two different papers to see the improvement.

The effect of the only implemented algorithm improvement (VIIRS cloud mask) has been addressed in our reply to comment (2) above.

 Later they mention needing a finer resolution surface albedo map, and there is also mention of the calibration causing some of the offset in the validation plots. Each of these issues is very interesting to another algorithm developer, like myself, or to potential users of the products. AMT is the right journal to present an analysis of these issues, and prove their consequence on the retrievals. Currently that analysis is missing.

In principle, as discussed in the manuscript, the identified AERONET-TropOMAER positive AOD bias (~0.2) could be the result of remaining calibration offset and/or issues with the coarse resolution of the currently used surface albedo data base. A calibration error will affect all AOD retrievals (independently of AOD magnitude) whereas a surface-albedo related error will impact retrieved low AOD values (up to ~ 0.5). At larger AOD's  surface-albedo-related effect become increasingly smaller. Specific conclusions regarding the magnitudes of these effects in TropOMAER are not yet available as we continue to investigate them. The discussion following the validation analysis includes these considerations.

(4) Slow down and present the details. I felt that there was a rush through the "boring" algorithm piece of the paper in order to get to the "exciting" demonstration with the big biomass burning events. There are many details left behind in the rush: There are many acronyms never properly introduced:

p.2 line 2 should put (SWIR) after shortwave infrared.

Done

P2 line 5. ESA and DLR?

Done

P2 line 28. Should put (ALH) after aerosol layer height

Done

P5 line 5. UVAI is never defined as an acronym, and worse, it is never defined as a product. Suddenly it is being shown in figures and being used as a fundamental part of the analysis.

This shortcoming has been addressed in the  added algorithm description section.

P6 line 25 SAM?

Stratospheric Aerosol Mass

P6 line 33. What are total mappers?

Nadir looking full daily coverage sensors (no longer in the discussion)

The concepts of Level 1 and Level 2 data are not explained (p2 line 5).

Done

Exactly what AERONET data are we looking at? Version 2 or 3? Levels 1.5 or 2? There is no explanation that AERONET AOD has a documented uncertainty of 0.02 in the UV, but that the SSA retrieval is a retrieval with much broader error bars. There is no explanation of why or how these 12 stations are selected, nor what the time range we are looking at.

Version 3 Level 2 data

(5) Provide more detail in the demonstration section. Figure 3 would benefit greatly by adding a swath just to the west of the swath shown. Right now there is a lot of description of fires and smoke in California, the Pacific Northwest and British Columbia, but none of those areas are shown in the figure. Only the areas downwind.

Added another orbit as suggested.

P6. Lines 1 to 6. Is this method here the manifestation of the ACA part of the TropOMI retrieval that is mentioned at the beginning? If so, then please make that clear. If it is a different method, then explain why the referenced ACA method is not used. If not, then is there any demonstration of the ACA TropOMI method? ACA is an important new addition to OMIAERUV, and should be highlighted or discussed if this is going to AMT.

It is the same. Stated in the manuscript.

P6 Line 10. The extinction-to-mass conversion is important. The appendix should be referenced here.

Done

P6 lines 13-16. Is there a physical basis for this? This is important, and how the UVAI AOD relationship relates to height, and especially to height in the stratosphere needs to be explained. Remember that UVAI jumps in suddenly with no introduction. It would be worthwhile to take the time to explain it, and some of the physics behind the whole interrelationship between height, AOD, UVAI and absorption. Maybe in Section 2?

For given values of ALH and AAE, UVAI increases rapidly with aerosol load up to AOD values in about the range 4-6 when it starts to saturate. At these large AOD's the aerosol absorption of Rayleigh scattered light peaks, and further UVAI enhancements are only possible for increased values of ALH and/or aerosol absorption exponent (AAE). Thus, for AOD values larger than about 6, and known or assumed AAE, the UVAI effectively becomes a measure of ALH. As suggested, this discussion has been included in section 2, where the UVAI concept is first introduced.

P6 line 25 to P7 line 2. A lot of numbers are given here and these are means with uncertainties surrounding them. The uncertainty is given at the end of±40%. It would be helpful to explain how the mean is derived (for what density) and what is the interplay between assumptions of density and uncertainty in height.

We meant uncertainty in AAE. ALH is given by CALIOP.
The uncertainty of the estimated stratospheric aerosol mass (SAM) is ±40% which represents the combined effect of uncertainties on assumed AAE (4.8±0.5) in the AOD retrieval, and the uncertainty in assumed aerosol density in the range 0.79 and 1.53 g-cm−3, which covers the range of values reported in the literature (Reid et al., 2005). For simplicity, we assume a midrange aerosol mass density value of 1.16 g-cm−3. These details are part of the discussion in the revised manuscript.

P7 lines 27-33. This is very interesting, but the figure doesn't really portray this information well. Figure 5 needs to become more informative.

> (6) All the captions need to more descriptive. Be sure to give details on specific data, be sure to describe what is shown in each panel, what wavelength is being shown, what temporal resolution is being plotted (fig. 5), what do each of the colors in the color bars represent. But in general a LOT more information needs to be in the figure captions.
>
> We assume the reviewer means fig 8.
> Figure 13 (previously Fig 8) shows calculated daily values of aerosol mass (in kilotons) from December 31, 2019 thru January 7, 2020, resulting from aerosols above 12 km, altitude used as a proxy of the tropopause height. Separate aerosol mass retrievals were carried out for cloud free (blue bars) and cloudy scenes (green bars), with the daily total stratospheric aerosol mass given as the sum of these two components (orange bars).

Suggestion: It occurred to me that this manuscript might fit a "letters" journal much better. Right now it is not too long. The authors would need to triage their figures down to 4. Perhaps Figs. 1, 3, 5 (with a bottom panel showing the difference between TropOMI and OMI) and 8. Then the very short description of the algorithm, evaluation and methods would be appropriate, and the purpose of the paper is NOT to describe TropOMAER, but to illustrate these biomass burning events. The point of the paper shifts from an "atmospheric measurement technique" to a better understanding of the Earth's atmospheric phenomena. GRL would be a possibility, but also ERL.

Thanks for the suggestion. We decided to stay with AMT

---

## Author Comment (AC2) · 16 Aug 2020

As a result of the review process, the manuscript has been modified significantly. Mayor changes are:

1) Section 2 of the paper has been extended to include a brief but detailed description of the TropOMAER algorithm. It includes a description of the UVAI calculation as well as a summary of the AOD/SSA retrieval process.

2) Section 3 on the validation of retrieval results using AERONET observations also changed considerably. The original validation analysis consisting of a direct validation of TROPOMI AOD results to AERONET observations at 12 sites was replaced with an approach that allows the separate evaluation of retrieved product improvement as a result of instrument enhancement and algorithmic improvement. AERONET observations 12 sites are used as an aggregate. A three way validation exercise is then carried out: 1) AERONET vs OMI, 2) AERONET vs TROPOMI using heritage (OMI) cloud mask, and 3) AERONET vs TROPOMI using VIIRS-based cloud mask. Inter-comparison for validations 1 and 2 highlights the effect of improved instrumental capabilities, whereas differences in validations 2 and 3 indicate retrieved product improvement due to algorithmic upgrades.

3) The revised paper (to be available soon after the submission of replies to reviewers' comments) contains 13 figures (five more than in the original version).

In the reply below the reviewer's comment is in black and our answer in blue.

Reply to Comments by Reviewer 2

This paper briefly introduces a TropOMI aerosol data set based on heritage OMI UV algorithms by the Torres group (OMAERUV and OMACA). This provides UV aerosol index (UVAI), aerosol optical depth (AOD), and single scattering albedo (SSA). A comparison of AOD and SSA against data from selected AERONET sites is presented, along with a few case studies of extreme events. The concept of the paper is in scope for AMT. The quality of language is good. The topic is important because OMI is ageing and TropOMI is the next generation of this type of sensor (OMPS on SNPP and JPSS has some aerosol capabilities but is in other ways worse than OMI).

However, honestly, the current paper feels more like a conference proceedings or an article for a Letters journal than a full scientific paper. It is brief and does not go into much detail. For a focused journal like AMT I think something much more technical is needed. Though I realise I am proposing a fair amount of work, I prefer that the authors expand this analysis rather than resubmit elsewhere, because I think a thorough accounting for TropOMI's capabilities for UV aerosol remote sensing is needed and is Interactive more or less missing from the literature. The authors are the right people to do this comment because they are the most expert with their data products. I know it is annoying when reviewers ask to do more work, but there is not enough content here to justify publication and I don't think that the article as written satisfies the scope a reader would reasonably expect. Case studies are one thing but by nature are typically unusual events and so looking at them may not give a representative picture of the data set as a whole. I recommend major revisions and would like to review the revision.

The paper has been significantly extended to address the issues raised in the review process.

My main suggestion for expansion is to give a detailed comparison between OMI and TropOMI results.

The original evaluation analysis involving AERONET-TROPOMI comparison of aerosol derived products have been converted into a three-way AERONET-TROPOMI-OMI over the same period.

OMI-TROPOMI results are compared for individual events as well as in terms of monthly averages for three representative regions as well as seasonal (summer) global averages.

Users familiar with OMI need to know whether we can use TropOMI for the same types of research, and to what extent the same caveats/biases are found. Right now this is not answered in a thorough way. One big advantage of TropOMI over OMI is the spatial resolution. I would expect that this is important because those cases where the UV technique works well (absorbing aerosols) are also often strong and heterogeneous events. So the finer spatial resolution might mean both (1) less cloud contamination and (2) better AOD/SSA retrievals, because top of atmosphere radiance is not linear in AOD, so by resolving more spatial structure you become less sensitive to sub-pixel variations. If this is true in practice, great. If not, this needs to be shown and understood. It is briefly discussed in Section 3.1 but not supported by the plots shown, only by briefly mentioning other references. Here are some suggestions for relevant analyses to include:

The revised version of the paper specifically addresses the issues addressed by the reviewer as explained below.

(1) Show global maps so we can see how similar the big picture looks from both sensors. In my view the time series in Figure 5 isn't sufficient here because both data sets are heavily spatially and temporally averaged in it.

Because of the so-called row anomaly of the OMI sensor that reduces OMI's daily coverage to about 50%, OMI-TROPOMI global daily maps are not the best way visual comparison. We show OMI-TROPOMI comparison on daily, monthly regional, and global seasonal temporal scales.

In Figure 1 of the revised version of the manuscript we show a comparison of OMI, NASA-TROPOMI and KNMI-TROPOMI UVAI on August 18 over North America. To our knowledge, except for UVAI, no other TROPOMI aerosol products are available.

Side-by-side maps of OMI and TROPOMI retrieved SSA and AOD for the same event are shown on Figure 8.

A two-year time series of monthly-averaged OMI and TROPOMI AOD and AAOD (absorbing aerosol optical depth) over three regions are shown on Figure 4.

OMI and TROPOMI summer 2018 seasonal global maps are compared in Fig 6, and a scatter plots of OMI TROPOMI UVAI monthly mean values is shown on Figure 7.

(2) Include OMI in some of the case studies (e.g. visual inspection of maps).

OMI graphics similar to the TROPOMI images have been added to the discussion of the 2018 California and Pacific northwest fires.

(3) OMI validation results could be presented alongside the TropOMI data. I know the validation has been published elsewhere but it will be clearer to the reader if plots are shown next to one another with the same axis range, etc.

The focus of the comparison to AERONET has changed from the narrowly focused AOD validation exercise in the original version of the paper, to an analysis of the instrumental and algorithmic differences throughout the use of independent ground-based observations. The combined AERONET data aggregate from observations the 12 sites, is compared to satellite observations as follows. An evaluation of instrument-related improvements is done by comparing AERONET measurements to three satellite-based data sets:1) OMAERUV, 2) TropOMAER with heritage (i.e., OMAERUV) cloud screening, and 3) TropOMAER with VIIRS cloud mask.

A comparative analysis of evaluations 1 and 2 shows the impact of enhanced instrumental capabilities, whereas the analysis of evaluations 2 and 3 highlights the effect of using the VIIRS cloud mask which is the only TropOMAER algorithmic modification.

(4) Directly plot (as a scatter density diagram) the AOD and/or UVAI from OMI and TropOMI, for collocated pixels (i.e. same scene, same time, similar geometry) at level 2 resolution. The orbits should overlap frequently. Then we can see if there's much scatter, if it's a straight line or not, etc. I don't know how much collocated data is needed to get a meaningful comparison – perhaps the case studies give enough, perhaps it has to be done on a month's worth of data. MODIS or VIIRS data could be useful for extra context (and filtering); I know and the manuscript mentions that the TropOMI orbit choice makes it possible to take advantage of SNPP VIIRS for e.g. cloud masking.

Because of the row anomaly the orbital overlap the reviewer describes is very cumbersome and time consuming. Figure 7 shows a scatter plot of seasonally averaged UVAI for the data mapped in Figure 6.

We believe the OMI-TROPOMI comparative analysis at daily, monthly regional, and seasonal temporal presented offers a complete analysis of the equivalence and compatibility of these two data sets. Additional comparisons involving other sensors are beyond the scope of this manuscript intended as a paper on first results of the ported algorithm and not yet a consolidated product.

The above comments and suggestions all apply (potentially) to the DSCOVR-EPIC sensor, too, although OMI is the more well-known and mature record so probably makes better sense to baseline against. Though I would certainly be happy to see a three-way (OMI, EPIC, TropOMI) comparison.

We will certainly carry additional comparison to other satellite products in the near future.

Other comments on the study are as follows:

Introduction or section 2.1: somewhere here it would be good to contrast TropOMI capabilities (e.g. spatial/spatial) with OMI and maybe TOMS and EPIC, since those are the main comparative products. The introduction mentions GOME and SCIAMACHY but those are less relevant since the authors' algorithms are from TOMS/OMI heritage and EPIC data are shown later. Maybe mention OMPS too as while a step backwards from OMI in terms of spatial resolution, it is used for UVAI and is the US operational follow-on for that. I know that there are TropOMI products in development on the Dutch side too – it's not clear to me whether those are public yet, but if so, there may be value in comparing and contrasting with those too.

The TOMS, EPIC and OMPS records are included in the discussion.

Section 2.2: if I understand correctly this section states that (1) there is a 5-10% calibration difference between OMI/OMPS and TropOMI in the relevant bands in the standard calibration, and (2) because of this the authors do their own vicarious calibration. Is that right? Either way, this could be worded a little more clearly. What is the difference between the sensors after the vicarious calibration?

The vicarious calibration brings the TROPOMI and OMI closer in measured reflectance terms as evidenced by the AOD validation presented here that shows overall consistency between the two records. The revised version of the manuscript contains an improved description of the vicarious calibration procedure.

Section 3: clear statements and references about AERONET data products and versions used need to be made. For example, I assume this is version 3 level 2.0 direct Sun (Giles et al AMT 2019) and inversions (Sinyuk et al AMT 2020). However this does not appear to be actually stated in the paper. If this was not the versions used, the analyses should be repeated using the latest data versions.

Yes, AERONET data version 3, level 2.0 was used. It has been clearly stated in the revised version of the paper.

Section 3.1: if the authors really believe that a relative uncertainty of 30% on TropOMI AOD is true, then by definition they should not be using linear least squares regression fits, because a relative uncertainty means that the assumption of constant variance of errors is broken. See for example standard statistics textbooks or web pages such as https://statisticsbyjim.com/regression/heteroscedasticity-regression/ . This issue could be addressed with weighted least squares. Ideally also the uncertainty on AERONET AOD (I think 0.02 in this spectral region) should be accounted for in the fitting. Also, if you expect a relative uncertainty then RMSE is not the best metric to be reporting since that is scale-dependent…others like relative RMSE would be more appropriate to quote instead/as well (and this would help tell you if it is really 30%). The statistical analysis here is not very appropriate. The authors may have used this type of analysis before but that does not mean it is ok to do something again if it is wrong.

TROPOMI's retrieval uncertainty is probably lower than the quoted 30% value. This is actually a conservative TOMS/OMI based estimate that includes the combined effect of the uncertainty on assumed aerosol layer height (smoke and dust layers)  and sub-pixel cloud contamination.  At TROPOMI's much finer spatial resolution the cloud contamination component should be significantly lower. Actual uncertainty is still to be determined pending remaining calibration issues as discussed in this manuscript. We appreciate the reviewer's observation on the appropriateness of using linear square regression (LQR) fits . LQR analysis have been used as a standard method of validating satellite AOD retrievals  The use of this common approach facilitates the relative comparison of the same physical parameter measured by large variety of sensors and retrieval algorithms.

The reported LQR parameters in this manuscript based on relatively small sample of observations are only intended to illustrate relative improvement in the accuracy of retrieved parameters associated with TROPOMI enhanced instrumental and algorithmic capabilities with respect to OMI. We do not expect the conclusion of our analysis to change if a more refined  fitting approach was used. This is by no means an exhaustive validation exercise of the TROPOMI record for which a lot more AERONET observations are needed.

Section 3.2: the authors use a 6 hour time window (3 hours each side) for the SSA comparison because morning/evening almucantar inversions have lower uncertainty than midday ones. The untested assumption here is that SSA does not vary much throughout the day. Ok, but version 3 also introduced hybrid scans which were specifically developed to solve this problem by sampling a larger air mass and scattering angle range during the middle of the day. This could be checked by using the hybrid inversions as well and seeing if you get the same results.

Hybrid scan availability is limited to specific sensor types. In general, reliable AERONET SSA retrievals are done for AOD (440 nm) > 0.40. That limitation significantly reduces the number of SSA measurements available for comparisons to satellite retrievals. Using hybrid scans only further reduces data available.

The hybrid scans are certainly useful to examine the issue of diurnal variability. We will consider using them in future specific validation efforts.

Also, an explanation is needed for how the authors split the data into the three aerosol type categories for Figure 2 and the discussion.

The aerosol typing is described in a new section of the paper that describes the algorithm as suggested by reviewer 1

Section 4: this feels like advertising. I agree that TropOMI results look impressive but (aside from a brief mention of AERONET AOD) there is no way to know how 'real' they are. This section feels like

something you might put on a webpage or brochure to attract attention to your new data set, rather than a detailed scientific analysis. I am not sure what is best to do here. For a journal like AMT I'd rather than space was devoted to more technical, large-scale comparisons. Perhaps this aspect could be split off for a Letters journal. Or, expanded with more context from meteorology and other (space or suborbital) data records and submitted separately to ACP. I know this is a joint special issue but the content still needs to match the journal. It does not really fit here, and there's not enough detail presented to consider this paper an authoritative reference for these case studies.

We disagree with the negative connotation of the term 'advertising' as used by the reviewer. As a matter of fact, this entire paper, not just section 4, as well as all science papers, are intended to introduce and advertise the availability of a new science products or ideas. That is the role of the scientific literature. The problem is when false advertisement takes place. Hopefully, the preceding three sections of the paper on algorithm description and evaluation of derived products give the reader some confidence to treat as 'real' the discussed practical applications of the derived products in section 4.

Figure 6 and associated text: I'm not sure that it makes sense to show the EPIC results on the left panel. That's a different sensor, different resolution, different observation geometry (backscatter for EPIC). UVAI is sensitive to all of these things. Also, what is the scaling referred to in the left panel? That is not mentioned in the paper.

Left panel Figure 6 has been excluded as it does not add much to the discussion without going into an additional explanation and description of the EPIC sensor. The EPIC application referred to in this paper is discussed in detail in the quoted literature.

I expect that the general point about the two events will still stand but it's not clear how much of the systematic difference (and scatter on the left panel) are a function of real differences in the smoke in the two events and how much is contributed by sensor differences. The paper is far too sparse in detail for a reader to judge, which makes the comparison less instructive.

Figure 6 left panel has been removed.

Figure 6 legend: is the black dot in the left panel legend (12 km) meant to be a black line like in the right panel? If so, formatting should be consistent. If not, the difference needs to be explained.

Figure 6 left panel has been removed.

Section 5: "The NASA TropOMAER aerosol algorithm is a modified version of the one applied to OMI observations." Wait, what? Section 2 describes the OMI approach but doesn't clearly state that there are modifications. What are these modifications, why were they made, what effect does this have on the results, and will they be back-ported to OMI? This all needs to be addressed in the paper.

Do not panic. The only modification is the use of the VIIRS cloud mask whose effect in retrieval results has been discussed.

---

## Author Comment (AC3) · 16 Aug 2020

As a result of the review process, the manuscript has been modified significantly. Mayor changes are:

1) Section 2 of the paper has been extended to include a brief but detailed description of the TropOMAER algorithm. It includes a description of the UVAI calculation as well as a summary of the AOD/SSA retrieval process.
2) Section 3 on the validation of retrieval results using AERONET observations also changed considerably. The original validation analysis consisting of a direct validation of TROPOMI AOD results to AERONET observations at 12 sites was replaced with an approach that allows the separate evaluation of retrieved product improvement as a result of instrument enhancement and algorithmic improvement. AERONET observations 12 sites are used as an aggregate. A three way validation exercise is then carried out: 1) AERONET vs OMI, 2) AERONET vs TROPOMI using heritage (OMI) cloud mask, and 3) AERONET vs TROPOMI using VIIRS-based cloud mask. Inter-comparison for validations 1 and 2 highlights the effect of improved instrumental capabilities, whereas differences in validations 2 and 3 indicate retrieved product improvement due to algorithmic upgrades.
3) The revised paper (to be available soon after the submission of replies to reviewers' comments) contains 13 figures (five more than in the original version).

In the reply below the reviewer's comment is in black and our answer in blue.

Reply to Comments by Reviewer 3

This paper presents NASA aerosol product for TROPOMI obtained with TropOMAER retrieval algorithm. In general, the manuscript is well-written, well-structured and demonstrates the possibilities of TropOMAER retrieval algorithm. First, the AOD and SSA products were evaluated using AERONET dataset for 12 representative sites. Then, the results of the algorithm application to a few important aerosol events were presented and total aerosol mass injection was estimated. There are few remarks regarding AOD and SSA validation against AERONET.

1. Figure 1 and Table 1 clearly indicate the presence of positive bias in TropOMAER AOD product at 380nm over all 12 representative sites. Authors already provided some guess about the origin of this bias and mention that this issue is under investigations. Nevertheless, since the retrieval is carried out at 388 nm, and reported also at 354 and 500 nm, presenting AOD validation results in the manuscript for two wavelengths (for example, 380 and 500 nm) would be very useful to address the bias issue.

The TropOMAER reported 354 and 500 nm AOD values are obtained by direct conversion from the retrieved 388 nm product that is based on the assumed spectral dependence of the aerosol models. We do not think the small wavelength difference between the AERONET 380 nm, and the satellite reported value at 388 nm explain the reported difference in the comparison. In regard to the evaluation at 500 nm, the added uncertainty of the reported AOD associated with the wavelength dependence would only make the interpretation of results more complicated. The suggestion, however, is very good and will be considered in upcoming evaluations of TropOMAER results.

2. One of the parameters of AOD evaluation is 30% matchup criteria. What is the origin of these criteria? Is AOD product with 30% uncertainty sufficient for trace gases retrieval? For example, GCOS requirements on AOD are much more strict: 0.03 or 10%.

TROPOMI's retrieval uncertainty is probably lower than the quoted 30% value. It is not, however, used as a matchup criterion. This value is actually a conservative TOMS/OMI-based estimate that includes the combined effect of the uncertainty on assumed aerosol layer height (smoke and dust layers) and sub-pixel cloud contamination. At TROPOMI's much finer spatial resolution the cloud contamination component should be significantly lower. Actual uncertainty is still to be determined pending remaining calibration issues as discussed in this manuscript.

3. The results of SSA validation show reasonable correspondence with AERONET. Nevertheless, Figure 2 clearly shows overestimation of SSA especially for absorbing aerosol when SSA from AERONET < 0.9. Is this related to the same issues providing positive bias in AOD? Is this SSA overestimation a demonstration of limitation of aerosol model used in TropOMAER algorithm? More discussions here are necessary.

The revised version of the paper includes parallel AERONET-OMI and AERONET-TROPOMI evaluations of both AOD and SSA products. The observed apparent overestimation of the satellite SSA values for desert dust aerosols is also present in the OMI comparisons (Figure 3a) and has been discussed in published literature (Jethva et al., 2014). Such overestimation, however, is not as clear in the presence of carbonaceous aerosols. The larger-than-AERONET desert dust SSA values (when AERONET < 0.9) are also observed in the TropOMAER evaluation for both the heritage (Figure 3b) and VIIRS (Figure 3c) cloud screening approaches. A smaller but observable similar effect is also apparent in the TROPOMI evaluation, suggesting a possible connection with lingering sensor calibration issues.

In general, I would recommend authors to reserve some space in the manuscript for discussions regarding identified issues in the retrieval. For example, the mentioned above issues for AOD and SSA retrieval as well as authors thoughts how to treat these issues would be highly appreciated by broad remote sensing community. These discussions would greatly increase the scientific strength of the paper.

These issues are discussed in the revised version of the manuscript.

---

## Author Response (AR2)

We are grateful to the reviewers' valuable feedback.

Reviewers' comments are reproduced in black. Our replies are in blue.

**Reply to Reviewer 1**

I reviewed the previous version of this paper. My main concern then was that it did not provide enough technical detail on the TROPOMI product and its performance relative to OMI to inform a data user. I also thought the case studies of recent major aerosol events weren't fleshed out enough for that to be the main focus. In this revision the authors have expanded the study significantly, especially in terms of TROPOMI-OMI comparability, which removes my main concern. This version has sufficient new content and aligns better with the scope of AMT; it is also clearer to read. In these respects it is a very good paper and will be of a lot of interest to the community.

We thank the reviewer for his/her helpful review.

Unfortunately the authors did not address a statistical problem (invalid use of linear least squares regression) I pointed out in my previous review. The data violate the assumptions made by this analysis technique. This concerns sections 3.1.2, Table 2, Figure 3, and some later discussion (including the Summary at the end of the paper). Here I will try to articulate more clearly why this is an issue. I therefore recommend further revisions to fix this.

The remedy is simple: just delete the lines, intercept, and slope, and the discussion. Removing it will not harm the paper. I appreciate the authors adding cautionary wording (page 7) not to over-interpret but it would still be better to delete these.

I don't believe it is responsible to publish bad statistics, especially when authors and editor are aware of the fact; it does nothing except inform people they can get away with it. I am open to a valid counter-argument to this but am yet to hear one; "it is common" (as here) is not a scientific argument to me. I am not trying to shut down the paper, it is a good paper other than this.

As an alternative, the authors could overplot binned median and standard deviation of error (or similar) as a function of AOD on figure 3 instead of the regression. It will be more informative as to the actual distribution of retrieval errors. We can look at the data, at the correlation and RMSE (which are not ideal but are less problematic diagnostics for the present purpose), and see TropOMI is better.

We have followed the reviewer's  suggestion and removed all discussion related to the linear fit analysis. As a result of this change, we have also removed the linear fit line from figure 3 and added to it the expected theoretical uncertainty associated with assumptions and aerosol layer height and cloud contamination. This explanation has been added to the paper.

The regression just muddles the issue as it invites the authors and readers to make an interpretation which is flawed because of the use of inappropriate statistics. As a case in point, Table 2 gives intercepts around +0.25 for Figures 3b and 3c. If you look at the data, it is clear that the point cloud of AOD up to about 0.5 is not pointing towards those being the actual intercepts if an AOD of 0 were measured. The true intercept looks smaller (but still positive). There are a small number of outliers pulling it up which are likely not reflective of the actual data. Regression amplifies these because the outliers are more extreme than the technique assumes. The position and torque of that cloud (AOD up to 0.5) may be different from that for higher AOD. So the relationship is not linear on aggregate. And as the authors note a relative uncertainty means those latter points shouldn't be weighted as heavily anyway. All of which is why you get an artificial high intercept and low slope.

We appreciate the reviewer's detailed explanation illustrating his/her point. We do fully agree with the referee that the initially reported linear least squares regression analysis is unduly driven by a number of outliers coming from a few sites.

Sure, TROPOMI calibration issues likely are real and cause a bias but it seems a stretch to imply this is causing an offset of +0.25 in low AOD. Figure 3a (for OMI) has a similar issue: regression intercept is +0.1 with again a small number of extreme outliers pulling it up. If you look at the OMI AOD, when AERONET AOD is low most of the time OMI is in fact around or below the 1:1 line. So what is this intercept telling us that is useful? Nothing, it is misleading us compared to if we look at the data. Yet these are the numbers highlighted in the paper's Conclusion. The regression adds nothing of value and hides information in a biased way. Just take a close look at Figure 3.

A detailed analysis of the comparison at individual sites, shows that most outliers in the range of low AOD (up to 0.5) shown in figures 3b and 3c come primarily from three sites: Banizoumbou, Beijing and Mongu. At these locations carbonaceous aerosols and sub-pixel size clouds co-exist, making cloud screening a particularly difficult task. This finding suggests that, as suggested by the referee, the initially reported high y-intercept and low slopes resulting from the linear least squares fit are driven by outliers at a few sites, and not entirely the result of severe calibrations issues that would show up at every location. To fully document this issue we have added, as an appendix, a figure that includes the scatter plots for each of the individual sites used in the analysis shown in Figure 3c.

I am not trying to be negative. I respect the authors' work a lot and they (here and elsewhere) do a very nice job getting the most out of spaceborne UV measurements. They continue to make improvements which enable people to do new and exciting science unavailable from other platforms. I just want bad statistics to stop being published when there is no need to.

Thanks. We certainly appreciate the feedback.

I had a couple of other small comments:

Figures 5, 6: How are standard deviations here calculated? I am not sure if this the standard deviation of all retrievals in the month, or between days from a daily average, or spatially from a monthly average, for example. This should be stated in text or caption. I ask as some of these (e.g. eastern US, Jan 2020 AOD) have a very high standard deviation and I am not sure if this is attributed to spatial variability across the region or an event causing temporal variability within that month, or something else.

The shown standard deviations are associated with both temporal and spatial variability. This is clarified in the figure caption in the revised version of the manuscript.

Page 11 lines 9-10: "where TROPOMI measured monthly average AOD in the vicinity of 1.0 0.9 are reported. downwind over the southeast" It looks like some text got cut off here as "1.0 0.9" does not make sense and then there is a loose sentence fragment.

The apparent incoherence of the alluded paragraph has been corrected.

**Reply to Reviewer 2**

This revised manuscript now reads like an AMT paper. I congratulate the authors for digging in to produce a much more informative paper. I believe now the manuscript is ready for publication with just a list of mostly technical corrections to make to the text.

We thank the reviewer for his/her helpful review.

I also make comments, but these should not be taken as requirements to be rectified, but more as suggestions for the authors to consider in either revising this paper or proceeding to the next one.

Corrections and comments.

p. 2 Line 12. Insert "an" to read "Per an established"
p. 2 Line 24. Insert a space between "section" and "3"
p. 3 Line 31. Insert a hyphen to read "TROPOMI-measured radiances"
p. 3 Line 31. Change "to" to "into" to read "input into a two-channel"

Suggested corrections above have been implemented p. 7 Lines 35-37. It may not be calibration. Could it be something particular to the 331 new points that TropOMI picks up that OMI misses?

It is certainly possible that any calibration offset may not be as large as suggested by the initially reported statistics. A detailed analysis of the comparison at individual sites, shows that most outliers in the range of low AOD (up to 0.5) shown in figures 3b and 3c come primarily from three sites: Banizoumbou, Beijing and Mongu. At these locations carbonaceous aerosols and sub-pixel size clouds co-exist, making cloud screening a particularly difficult task. This finding suggests that the initially reported high y-intercept and low slopes resulting from the linear least squares fit are heavily weighted to sub-pixel cloud contamination at a few sites  and not entirely the result of  calibrations issues that would show up at every location.  This is clearly shown in a new appendix added to the manuscript that includes the scatter plots for each of the individual sites used in the analysis shown in Figure 3c.

As suggested by other referee, in the revised version of the manuscript we have excluded the linear fit and, instead, focused  the discussion of the validation analysis in terms of correlation coefficients and root mean square error.  Improvements of the currently used cloud masking scheme is mentioned in the discussion section as an area where additional work is necessary.

p. 8 Line 16. Re-write to read "Because AERONET SSA derived from almucantar scans is considered unreliable near noon (Dubovik et al., 2002) when satellite overpass occurs, "
p. 8 Line 21. Insert "the" to read "Although the Version 3"
p. 8 Line 22. Delete "over" and insert "a" to read "covering a wider range"
p. 8 Line 23. Replace "is" with "are" to read "sensors are capable"
p. 8 Line 30. Replace "Table 2" with "Table 3"

Suggested corrections above have been implemented p. 9 Lines 22 – 24. I wouldn't be so sure that there is a cancellation of error or even that AAOD shows a closer agreement than AOD. Did you expect that AAOD would have a difference of 0.2 when the magnitude of AAOD is only 0.10? Differences would have to be smaller than AOD because magnitudes of AAOD are smaller than AOD. The relative difference in the AAOD is basically the same percentage as the relative difference in AOD.

As correctly pointed out by the reviewer a total cancellation of errors is not possible. Our argument if for the existence of a  small *partial* error cancellation.

p. 9 Line 31. Should read "northwest India"
p. 10 Line 6. Insert "the" to read "improved the near UV"
p. 10 Line 22. Insert "the" to read "to the California"

Suggested corrections above have been implemented p. 10 Section 4.1. What wavelength is this AOD? Also both sensors miss retrieving AOD in the core of the plume. Out of range of the retrieval? Masked for cloud?

Gaps are the result of out of range retrieval conditions.

p. 11 Line 9. Rewrite to read "where TROPOMI-measured monthly average AOD in the range of 0.9 to 1.0 are reported."

Done.

p. 11 Section 4.2. Do you want to give a reference for the media attention?

A reference to  Hughes, R.: Amazon Fires: What is the latest in Brazil? BBC News, October 11, 2019, has been added.

 Do you want to say something about TROPOMI continuing the OMI time series? Because if not, why is this in this paper? Also, I notice that there are differences in the time series between the two sensors. These differences are not large enough to question the ability to recognize big years from small years, but they are differences with respect to quantifying amounts.

Following the referee's suggestion the following paragraph has been added to the discussion: 'Figure 11 shows the time series of monthly average OMI 388nm AOD over the region for the last 15 years, along with the overlapping TROPOMI AOD observations over the last two years, illustrating the importance of the continuity of the longterm record. Although, as discussed earlier, there are small differences in the time series between the two sensors, these differences are not large enough to question the ability to recognize years with large seasonal events from years with comparably reduced biomass burning activity'.

p. 11 Line 22. Make "region" plural to read "over regions up north"
p. 11 Line 28. Change to read "including species that were near extinction before the fire"
p. 11 Line 30. Change to read "aerosols into the Southern Hemisphere"
p. 12 Line 8. Add a comma (this one is important) to read "provided ALH information, and assumed AAE value"

Suggested corrections above have been implemented p. 12 paragraph from Line 25 to p. 13 Line 2. Some things are unclear to me. In equation A-1, the sum is over "the total area covered by the aerosol plume" meanwhile there is a parameter "A" in equation A-1. The "A" is the effective geographical area with retrieved stratospheric AOD. Are these the same thing? Or is "A" the area covered by a single pixel of the retrieval? Is the total area of the plume some sort of latitude longitude box, or the total area defined by whether or not there are aerosols determined to be above 12 km within some sort of latitude longitude box?

"A" is the area of each 0.25°x0.25° lat.-lon. grid. Only AOD retrievals for pixels inside the grid with ALH determined to be above 12 km are included in the mass calculation. This clarification has been added to both the main text of the manuscript and to Apprendix-A of the revised manuscript.

p. 12, more on the same paragraph. The statement about dilution is confusing. "spreads the aerosol layer horizontally and thins it out". Does this mean that the aerosol passes out of the area in the horizontal? If it were only a matter of spreading out horizontally but staying within the same domain, the total mass in the domain would be the same. Concentration would decrease but total mass is the same. It seems to me that what is happening is that the aerosol is falling below 12 km and thinning out because of deposition of some kind. I see it as a vertical issue not a horizontal one.

The reviewer brings up a good point. We believed the observed mass decrease is a combination of both aerosol detectability as well as possible aerosol deposition. The observed stratospheric aerosol mass decrease is likely due to the combined effect of dilution processes, that spread the aerosol layer horizontally and thins it out to extremely low AOD values beyond the sensor's sensitivity to the total AOD column measurement, as well as aerosol deposition bringing it down to lower than 12 km and, therefore, no longer included in the mass calculation. This explanation has been added to the manuscript.

p. 13 Lines 35-36. I did not understand this sentence. Is "exacerbated" the correct word here?

We replaced "exacerbated" with "over-estimated".

p. 14 Line 5. All through the paper there is an assumption that we know what wavelength is being discussed. It would be helpful occasionally, including here in the Summary, to say "AOD at XXXX" and give the wavelength.
p. 14 Line 20. Remove "presence" to read "levels of carbonaceous aerosol were detected in 2019"
p. 14 Line 26. Replace "in" with "into" to read "carbonaceous aerosols into the Southern"
p. 14 Line 28. Make "layers" singular to read "a distinct high-altitude aerosol layer near 12 km"
p. 14 Line 29. Add hyphen to read "TROPOMI-retrieved"
p. 14 Line 31. Replace "in" with "into" to read "injected into the stratosphere"
p. 23 Figure 2 caption. Please state the wavelength
p. 24 Figure 3 caption. Add at the end "See text for details and Table 2 for linear regression statistics." Also please state the wavelength.

Suggested corrections above have been implemented p. 25 Figure 4. With TropOMI you start to see a real separation in SSA by aerosol type that you don't see in OMI. Calibration is the easy explanation for the biases, but for the separation by type? Is it the additional data points? I wonder about the models used to move the SSA to 440 nm from 388 nm.

These are  good points. At this time we do not have a unique explanation. These issues will be examined in future  detailed  analyses.

p. 26 Figure 5 caption. Put spaces between "in" and "red; "in" and "blue"
p. 26 Figure 5 caption. Please state the wavelength
p. 30 Figure 9 caption. Please state the wavelength
p. 31 Figure 10 caption. Please state the wavelength
p. 32 Figure 11 caption. Please state the wavelength
p. 34 Figure 13 caption. Please state the wavelength

Suggested corrections above have been implemented.

[revised manuscript text omitted]